# Inhibiting cholesterol synthesis halts rhabdomyosarcoma growth via ER stress and cell cycle arrest

Nebeyu Yosef Gizaw [ID][1], Kalle Kolari [ID][2], Pauliina Kallio [ID][3], Kari Alitalo [ID][3,4] & Riikka Kivelä [ID][1,2,4 ✉]

## Abstract

**Rhabdomyosarcoma (RMS) is the most common pediatric soft tissue sarcoma, with poor outcomes in high-risk and relapsed patients. Here, we identify de novo cholesterol biosynthesis as a critical metabolic vulnerability in RMS. The transcription factor PROX1, previously implicated in RMS growth, acts as an upstream regulator of cholesterol biosynthesis, promoting expression of key pathway genes. Inhibition of cholesterol biosynthesis, either genetically or pharmacologically, impaired RMS cell proliferation, caused a broad halt of cell cycle progression, and activated ER stress-mediated apoptosis through the PERK–ATF4–CHOP axis. Notably, RMS cells could not be rescued by exogenous LDL cholesterol, indicating a unique reliance on endogenous cholesterol production, whereas normal cells, including myoblasts and astrocytes, largely relied on extracellular cholesterol uptake. Clinical and single-cell RNA-seq analyses further revealed that high expression of cholesterol biosynthesis genes correlate with poor survival and enrichment of cell cycle-related gene signatures across RMS subtypes. Together, these findings mechanistically link cholesterol biosynthesis to proliferative signaling and ER stress response in RMS and highlight this pathway as a promising, non-redundant therapeutic target.**

**Keywords** Drug Repurposing; Mevalonate Pathway; Pediatric Cancer; Sarcoma; Statins
**Subject Categories** Cancer; Metabolism; Musculoskeletal System

## Introduction

Rhabdomyosarcoma (RMS), an aggressive childhood cancer with poor prognosis, accounts for approximately 7% of all pediatric cancers and about half of all soft tissue sarcomas in children (Kashi et al, 2015). RMS tumors exhibit features of skeletal muscle and are believed to originate from mesenchymal cell precursors that have failed to differentiate or have developed abnormally along the skeletal muscle lineage (Hettmer and Wagers, 2010). The two major histological subtypes of RMS are embryonal RMS (ERMS) and alveolar RMS (ARMS), each driven by distinct mechanisms (Newton et al, 1988). The ARMS variant, which generally has a worse prognosis, harbors pathogenic chromosomal translocations in about 80% of cases, resulting in the expression of PAX3-FOXO1 or PAX7-FOXO1 fusion proteins (Davis et al, 1994). The remaining 20% of ARMS tumors lack these translocations and are classified as fusion-negative, with outcomes similar to those of ERMS. Consequently, RMS is often categorized in clinical settings as either fusion-positive (FP-RMS) or fusion-negative (FN-RMS) (Williamson et al, 2010). Treatment for RMS involves a multi-disciplinary approach, including chemotherapy combined with local control through surgical resection and/or radiation therapy (Yohe et al, 2019). However, the five-year event-free survival rate for patients with metastatic disease at diagnosis remains under 30%, and those with relapsed disease face similarly poor outcomes. Over the past three decades, neither the survival rates nor the treatment side effects for high-risk RMS have significantly improved (Kashi et al, 2015). Thus, improving these survival rates hinges on identifying clinically effective agents targeting RMS vulnerabilities.

Here, we investigated the molecular mechanisms driving RMS tumorigenesis. Our previous work showed that the transcription factor Prospero homeobox 1 (PROX1) is highly expressed in both FN-RMS and FP-RMS and acts as an essential regulator of myogenic and tumorigenic properties of RMS (Gizaw et al, 2022).

Through immunohistochemistry and RNA sequencing (RNA-Seq) analysis, we found that PROX1 regulates genes associated with cholesterol biosynthesis in both RMS subtypes. Deregulated lipid and cholesterol homeostasis often drive tumorigenesis and cancer progression (Snaebjornsson et al, 2020) and elevated expression of the cholesterol-biosynthesis pathway strongly correlates with poor prognosis in many solid tumors (Cai et al, 2019). This link is unsurprising, as cholesterol is crucial for cell membrane structure, and rapidly proliferating cancer cells require high cholesterol levels for membrane biogenesis and other functions (Duan et al, 2022; Huang et al, 2020). The mevalonate-cholesterol biosynthesis (MVA) pathway is a complex biochemical route that produces

[1]Stem Cells and Metabolism Research Program, Research Programs Unit, Faculty of Medicine, University of Helsinki, 00014 Helsinki, Finland. [2]Faculty of Sport and Health Sciences, University of Jyväskylä, 40700 Jyväskylä, Finland. [3]Translational Cancer Medicine Research Program, Research Programs Unit, Faculty of Medicine, University of Helsinki, 00014 Helsinki, Finland. [4]Wihuri Research Institute, 00290 Helsinki, Finland. ✉E-mail: riikka.kivela@helsinki.fi

several critical end-products, including cholesterol, isoprenoids, dolichol, ubiquinone, and isopentenyl adenine (Goldstein and Brown, 1990). Additionally, reprogramming of lipid metabolism can influence other crucial processes involved in cancer progression, including endoplasmic reticulum (ER) stress response and apoptosis (Snaebjornsson et al, 2020). Statins, which inhibit HMGCR, the first rate-limiting enzyme of the MVA pathway, have been demonstrated to reduce growth and increase apoptosis in various cancers both in vitro and in vivo (Jiang et al, 2014; Longo et al, 2020). Despite these insights, metabolic reprogramming in RMS, especially that of lipid and cholesterol pathways, is poorly understood and remains an untapped therapeutic target.

In the current study, we show that cholesterol biosynthesis is highly induced in the majority of human RMS cases. Inhibition of cholesterol biosynthesis resulted in decreased cell proliferation and colony formation in vitro. Additionally, cholesterol biosynthesis was found to be essential for the growth of tumor xenografts following the engraftment of human RMS into immune-compromised mice. Silencing of DHCR7, the last enzyme in the metabolic pathway regulating cholesterol biosynthesis, induced ER stress and ER stress-mediated intrinsic apoptosis in RMS cells and tumors. These findings suggest that targeting cholesterol biosynthesis could be a promising therapeutic strategy for RMS.

## Results

### Cholesterol biosynthesis is elevated in RMS and regulated by PROX1

Our previous studies have demonstrated that PROX1 plays a regulatory role in the myogenic phenotype in both skeletal myoblasts and rhabdomyosarcoma (RMS) cells (Gizaw et al, 2022; Kivela et al, 2016). Strikingly, increased PROX1 expression induces terminal differentiation in myoblasts, whereas in RMS, it promotes proliferation. To identify the core transcriptional program induced by PROX1 in RMS cells but not in healthy myoblasts, we conducted RNA-seq analysis on PROX1-silenced RD cells (FN-RMS), KLHEL1 cells (FP-RMS), and healthy human myoblasts (Fig. 1A). The silencing efficiency in each cell type is indicated by log2 fold change in the figure. Gene Ontology (GO) analysis of the 436 transcripts downregulated in both RMS cell lines, but not in the healthy myoblasts, revealed that cholesterol biosynthesis was among the most highly enriched repressed pathways (Fig. 1B). Of note, GO analysis of the 225 genes commonly downregulated across all three cell lines did not identify any significantly enriched biological processes but showed enrichment for cellular component terms related to the mitochondrion and mitochondrial matrix. Consistently, gene-set enrichment analysis (GSEA) clearly indicated that the hallmarks of the cholesterol biosynthesis pathway were significantly affected by PROX1 silencing in both RD and KLHEL1 cells (Fig. 1C). Importantly, in agreement with the observed effects on gene expression, PROX1 silencing significantly reduced the cellular cholesterol content of the RMS cells (Fig. 1D). Further analysis using previously published gene expression data from primary RMS patient samples (GEO: GSE108022) (Shern et al, 2014) revealed a significant upregulation of genes associated with cholesterol

biosynthesis in both FN- and FP-RMS compared to healthy skeletal muscle (Fig. 1E). Similarly, qRT-PCR analysis showed that both FN-RMS (RD) and FP-RMS (KLHEL1 and RH30) cells express significantly higher levels of cholesterol biosynthesis genes compared to healthy myoblasts (Fig. 1F).

Next, we analyzed human RMS tumor samples obtained from the Helsinki Biobank. Notably, immunohistochemical analysis showed that both PROX1 and the cholesterol synthesis enzyme DHCR7 were highly expressed in the tumor tissue compared to the adjacent healthy muscle (Fig. 1G).

### Inhibition of cholesterol biosynthesis suppresses RMS cell growth and survival

To evaluate the significance of cholesterol biosynthesis in RMS, we first silenced HMG CoA reductase (HMGCR), the first rate-limiting enzyme in the mevalonate–cholesterol biosynthesis pathway, in both the RD (FN-RMS) and KLHEL1 (FP-RMS) cell lines by using two independent lentiviral short hairpin RNAs (shRNAs) (Fig. EV1A,E). Live cell imaging and analysis revealed that the proliferation of RD and KLHEL1 cells was markedly inhibited by HMGCR silencing (Fig. EV1B,F). To assess whether HMGCR silencing affected the clonogenic potential of RD and KLHEL1 cells, we conducted colony formation assays, which revealed a remarkable reduction in both the number and size of colonies formed (Fig. EV1C,D,G,H). In line with the genetic experiments, treatment with the HMGCR inhibitor lovastatin, a cholesterol-lowering drug that slows down cholesterol biosynthesis, also significantly reduced RMS cell proliferation (Fig. EV1G,H).

The mevalonate–cholesterol biosynthesis pathway is critical not only for cholesterol production but also for the synthesis of other metabolites such as isoprenoids, which have been implicated in tumor progression (Fig. 2A; ref (Goldstein and Brown, 1990)). To elucidate whether it is indeed cholesterol or other upstream metabolites in the pathway that are essential for RMS growth, we specifically targeted the last enzyme in the pathway, DHCR7, using three independent shRNAs (Fig. 2B,G). DHCR7 silencing completely halted the proliferation of RD and KLHEL1 cells (Fig. 2C,H), and significantly reduced their colony formation capacity (Fig. 2D,E,I,J). Interestingly, the defective proliferation was not rescued by supplementing the culture media with LDL cholesterol, indicating that RMS cells are highly dependent on the de novo cholesterol synthesis (Fig. EV2A,B). Additionally, DHCR7-silenced RD and KLHEL1 cell cultures exhibited more apoptotic cells than shSCR-transduced controls (Fig. 2F,K). Consistent with the effects of DHCR7 gene silencing, the DHCR7 specific inhibitor AY9944 inhibited proliferation of RD and KLHEL1 cells in a dose-dependent manner (Fig. 2L,N). Furthermore, treatment with AY9944 compromised the survival of RD and KLHEL1 cells, as evidenced by pronounced apoptosis, measured by the activation of caspase 3/7 (Fig. 2M,O). Importantly, inhibition of DHCR7 by AY9944 also impaired proliferation and induced apoptosis in another FP-RMS cell line RH30, wheras primary human myoblasts and immortal human astrocytes (INHA) were not affected (Figs. EV2C,D and 2P–S). These data indicate that de novo cholesterol biosynthesis is essential for the proliferation, viability, and clonogenic capacity of RMS cells, irrespective of the RMS tumor subtype.

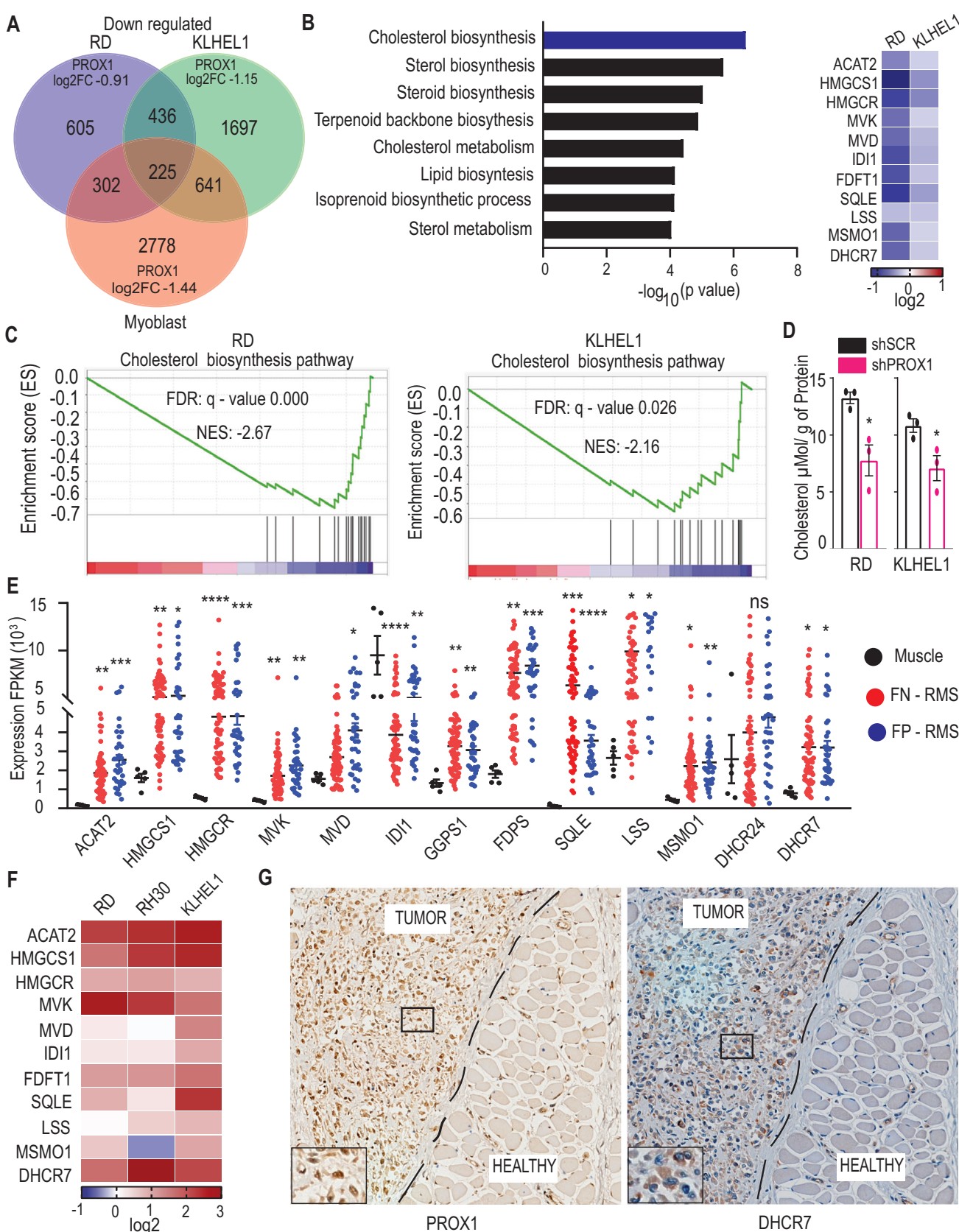

**Figure 1. Cholesterol biosynthesis is elevated in rhabdomyosarcoma (RMS).**

(A) Venn diagram illustrating the number of significantly downregulated genes identified through RNA-seq analysis following PROX1 silencing in RMS cell lines RD and KLHEL1 and compared to healthy human myoblasts. (B) Gene ontology analysis of the 436 common genes downregulated upon PROX1 silencing in both RD and KLHEL1 cells but not in human myoblasts. The heatmap shows downregulated cholesterol biosynthesis genes in both cell lines. (C) Gene Set Enrichment Analysis (GSEA) demonstrates the consistent repression of genes involved in cholesterol biosynthesis pathway upon PROX1 silencing in RD cells (left) and in KLHEL1 cells (right). NES denotes normalized enrichment score, and FDR indicates false-discovery rate. (D) Quantification of total cellular cholesterol content in shSCR and shPROX1 treated RD cells (left) and KLHEL1 cells (right) at 3 days post-transduction. The data were analyzed following organic extraction and normalization to protein levels. Data were presented as mean ± SEM; $N = 3$ biological replicates. Statistical analysis: unpaired $t$-test (two-tailed). Significance: $*P < 0.05$. (E) Normalized mRNA expression (FPKM) of genes involved in cholesterol biosynthesis in normal muscles, fusion-negative RMS (FN-RMS), and fusion-positive RMS (FP-RMS) tumor samples in a previously published RNA-seq dataset (GSE108022, FDR < 0.05). Each data point represents an individual tumor sample. The horizontal line indicates the mean expression and asterisks denote significance ($*p < 0.05$, $**p < 0.01$, $***p < 0.001$, $****p < 0.0001$; n.s., not significant, Student's t-test). (F) A heatmap showing mRNA expression of cholesterol biosynthesis genes in RD, RH30, and KLHEL1 cells relative to human primary myoblasts, analyzed by qRT-PCR ($n = 3$ per group). (G) Immunohistochemical staining of PROX1 (left) and DHCR7 (right) in clinical RMS tumor samples obtained from the Helsinki Biobank. The tumors exhibit robust nuclear PROX1 expression and cytoplasmic DHCR7 expression compared to adjacent healthy muscle tissue (tumor border marked with a dashed line). Exact $p$ values are reported in Table EV2. Source data are available online for this figure.

## Cholesterol biosynthesis is essential for the growth of RMS tumor xenografts

We next analyzed the tumor-propagating potential of cholesterol biosynthesis in vivo by engrafting control and DHCR7-silenced RD (FN-RMS, silencing efficiency ~75%) and KLHEL1 (FP-RMS, silencing efficiency ~80%) cells into the left and right flanks of female NOD/SCID/IL2rg mice (Fig. EV3A,B). Serial tumor volume measurements demonstrated significant growth inhibition in the DHCR7-silenced tumors compared to the shSCR-transduced control tumors (Fig. 3A). Tumors derived from the DHCR7-silenced cells were notably smaller and lower in weight than their respective controls (Fig. 3B,C). Hematoxylin and eosin (H&E)-stained tumor sections from DHCR7-silenced RD and KLHEL1 cells exhibited also reduced cell density compared to control tumors (Figs. 3D,H,L and EV3C,G,K). Immunofluorescence staining for Ki67 and cleaved caspase 3 (CC3) revealed decreased cell proliferation (Figs. 3E,I,I and EV3D,H,L) and increased apoptosis (Figs. 3F,J,N and EV3E, I, M) in the DHCR7-silenced tumors. DHCR7 RNA and immunohistochemistry (IHC) analyses confirmed lower DHCR7 expression in shDHCR7-derived tumors than in control tumors during tumor development (Figs. 3G,K,O and EV3F,J,N). These findings provided compelling evidence that cholesterol biosynthesis is indispensable for RMS tumor growth in vivo. However, based on these results it is not possible to distinguish whether DHCR7 silencing affected mainly tumor engraftment or tumor proliferation and growth.

## Blocking cholesterol biosynthesis inhibits RMS cell cycle progression and activates ER stress-triggered apoptosis

To elucidate global changes in gene expression in response to cholesterol inhibition in RMS tumors, we conducted whole transcriptome analysis on DHCR7-silenced and control RD cells. Principal component analysis (PCA) and hierarchical clustering of the RNA-Seq data revealed that the DHCR7-silenced and control cells formed distinct groups (Fig. 4A,B). Analysis of the differentially expressed genes (DEGs), using the statistical significance cutoff at the false discovery rate (FDR) of <0.05 and biological significance cutoff of ≥1.5-fold, identified 1961 upregulated and 2698 downregulated genes in the shDHCR7 cells (Fig. 4C). Among the top 10 most downregulated genes were the proliferation marker MKi67 and the transcription factor E2F2

(Dataset EV1), both of which play critical roles in regulating cell proliferation. Gene ontology analysis further highlighted cell division as one of the most significantly enriched biological processes among the downregulated DEGs (Fig. EV4A). The E2F family of transcription factors (E2F1, E2F2, and E2F3) are well-known activators of cell cycle progression, directly controlling the expression of key regulators such as cyclin E and CDC6 (Cam and Dynlacht, 2003; Chen et al, 2009; Helin, 1998). In agreement with this, Gene Set Enrichment Analysis (GSEA) revealed a broad suppression of E2F target genes, along with multiple genes involved in distinct phases of cell cycle progression (Figs. 4D–I and EV4B–D). Consistent with these transcriptomic findings, DHCR7 silencing markedly reduced cyclin D1 protein levels in all three RMS cell lines tested, representing both fusion-negative and fusion-positive subtypes (Fig. 4J). qPCR analysis further validated the downregulation of cyclin E and CDC6 expression upon DHCR7 silencing (Fig. 4K). Flow cytometry analysis in DHCR7 -silenced RD and RH30 cells demonstrated an increase in the proportion of cells in the G2/M phase, accompanied by a corresponding decrease in the G0/G1 and S phases (Fig. 4L). These findings suggest that inhibition of cholesterol biosynthesis abrogates cell cycle progression and induces cell cycle arrest predominantly at the G2/M phase, by modulating the expression of key cell cycle-related genes.

DHCR7 silencing also increased several key transcripts of the cholesterol biosynthesis pathway (Figs. 5A and EV4E), likely as a compensatory mechanism to increase cholesterol synthesis. This effect was also evident at the protein level, as DHCR7 silencing induced HMGCR expression in both RD and KLHEL1 cells (Fig. EV4F). Gene set enrichment analysis of the upregulated genes revealed strong enrichment of genes involved in the endoplasmic reticulum (ER) unfolded protein response (UPR) (Figs. 5B and EV4E). UPR is typically initiated and regulated by three ER sensors: inositol-requiring enzyme 1 (IRE1), PKR-like ER kinase (PERK), and activating transcription factor 6 (ATF6) (Chen et al, 2023; Lin et al, 2008). Our data demonstrated that transcripts linked to these pathways were increased following DHCR7 silencing (Dataset EV2). Notably, GSEA also revealed activation of the intrinsic apoptosis signaling pathway activated during prolonged ER stress (Fig. 5C). Consistently with this, our data showed significant upregulation of transcripts of the PERK branch of the UPR, which is known to elicit pro-apoptotic effects (Szegezdi et al, 2006) (Fig. 5D). Among these we found the activating transcription factor-4 (ATF4), which induces the

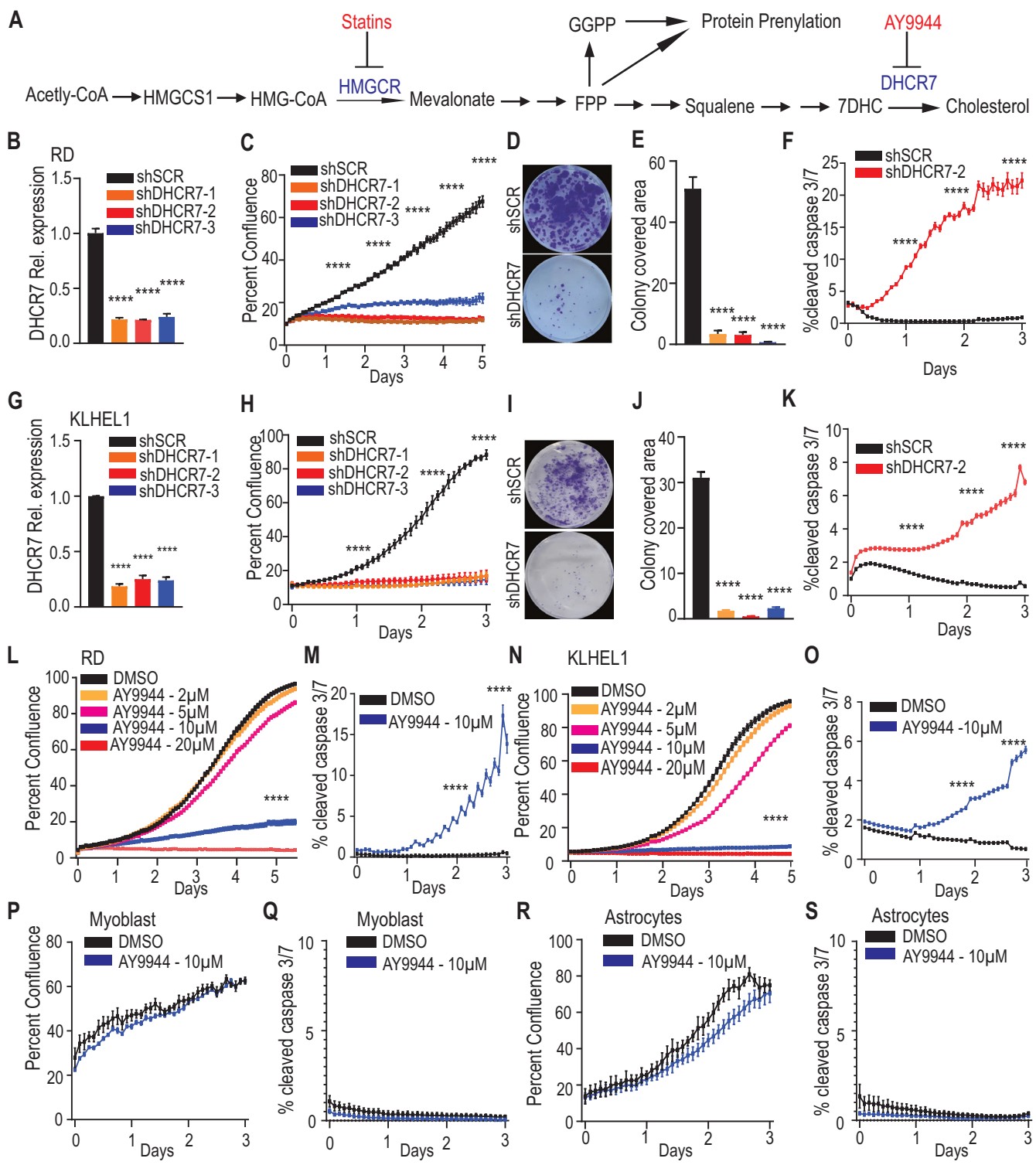

expression of C/EBP-homologous protein (CHOP), a regulator of pro-apoptotic genes during ER stress. Also ATF4 and its target genes, including CHOP and the CHOP-induced pro-apoptotic gene GADD34, were significantly upregulated by DHCR7 silencing (Fig. 5E and Dataset EV2). Upregulation of these PERK-related transcripts was also observed upon silencing DHCR7 in KLHEL1

and RH30 cells (Fig. 5F,G). Since PERK enhances ATF4 expression by phosphorylating its only known target, eukaryotic translation initiation factor 2α (eIF2α), we analyzed the phosphorylation levels of eIF2α and found increased phosphorylation of eIF2α in both RD and KLHEL1 cells following DHCR7 knockdown. This was accompanied by elevated ATF4 and CHOP protein expression

**Figure 2. Inhibition of cholesterol biosynthesis suppresses RMS cell proliferation and survival.**

(A) Schematic of cholesterol biosynthesis pathway starting from acetyl-CoA. HMGCR catalyzes the first rate-limiting step, and DHCR7 is the final enzyme in the pathway. (B) Quantitative PCR analysis of DHCR7 mRNA expression in RD cells following shSCR (control) and shDHCR7 transduction with three independent shRNA constructs. Data were presented as mean ± SEM; $N = 3$ biological replicates. Statistical analysis: one-way ANOVA analysis followed by Dunnett's multiple comparison test. Significance: ****$P < 0.0001$. (C) Cell proliferation assessed by IncuCyte live cell imaging for stably expressing shSCR or shDHCR7 RD cells utilizing three different silencing constructs. Data were presented as mean ± SEM; $N = 3$ biological replicates. Statistical analysis: one-way ANOVA analysis followed by Dunnett's multiple comparison test. Significance: ****$P < 0.0001$. (D, E) Colony formation assay and corresponding quantification in RD cells. Data were presented as mean ± SEM; $N = 3$ biological replicates. Statistical analysis: one-way ANOVA analysis followed by Dunnett's multiple comparison test. Significance: ****$P < 0.0001$. (F) Quantification of cleaved caspase 3/7 activity in control and DHCR7-silenced RD cells using a fluorescent reporter and live imaging. Data were presented as mean ± SEM; $N = 3$ biological replicates. Statistical analysis: unpaired $t$-test (two-tailed). Significance: ****$P < 0.0001$. (G) qPCR analysis of DHCR7 mRNA expression in KLHEL1 cells following shSCR (control) and shDHCR7 transduction with three independent shRNA constructs. Data were presented as mean ± SEM; $N = 3$ biological replicates. Statistical analysis: one-way ANOVA analysis followed by Dunnett's multiple comparison test. Significance: ****$P < 0.0001$. (H) Cell proliferation based on IncuCyte live cell imaging for stably expressing shSCR or shDHCR7 transduction with three independent shRNA constructs in KLHEL1 cells. Data are presented as mean ± SEM; $N = 5$ biological replicates. Statistical analysis: one-way ANOVA analysis followed by Dunnett's multiple comparison test. Significance: ****$P < 0.0001$. (I, J) Colony formation assay and corresponding quantification in KLHEL1 cells. Data are presented as mean ± SEM; $N = 3$ biological replicates. Statistical analysis: one-way ANOVA analysis followed by Dunnett's multiple comparison test. Significance: ****$P < 0.0001$. (K) Caspase 3/7 activity in control and PROX1-silenced KLHEL1 cells evaluated using a fluorescent reporter. Data are presented as mean ± SEM; $N = 3$ biological replicates. Statistical analysis: unpaired $t$-test (two-tailed). Significance: ****$P < 0.0001$. (L) Growth curves of RD cells treated with different concentrations of DHCR7 inhibitor (AY9944) and analyzed by live cell imaging. Data are presented as mean ± SEM; $N = 6$ biological replicates. Statistical analysis: one-way ANOVA analysis followed by Dunnett's multiple comparison test. Significance: ****$P < 0.0001$. (M) Caspase 3/7 activity in dimethyl sulfoxide (DMSO) control and DHCR7 inhibitor-treated RD cells. Data are presented as mean ± SEM; $N = 6$ biological replicates. Statistical analysis: unpaired $t$-test (two-tailed). Significance: ****$P < 0.0001$. (N) Growth curves of KLHEL1 cells treated with indicated concentrations of DHCR7 inhibitor (AY9944) and analyzed by live cell imaging. Data are presented as mean ± SEM; $N = 6$ biological replicates. Statistical analysis: unpaired $t$-test (two-tailed). Significance: ****$P < 0.0001$. (O) Caspase 3/7 activity in DMSO and DHCR7 inhibitor-treated KLHEL1 cells was assessed using a fluorescent reporter. Data are presented as mean ± SEM; $N = 6$ biological replicates. Statistical analysis: unpaired $t$-test (two-tailed). Significance: ****$P < 0.0001$. (O) Growth curves of KLHEL1 cells treated with indicated concentrations of DHCR7 inhibitor (AY9944) and analyzed by live cell imaging. Data are presented as mean ± SEM; $N = 6$ biological replicates. Statistical analysis: one-way ANOVA analysis followed by Dunnett's multiple comparison test. Significance: *$P < 0.05$, ****$P < 0.0001$. (P) Growth curves of primary human myoblast cells treated with DHCR7 inhibitor (AY9944) and analyzed by live cell imaging. Data are presented as mean ± SEM; $N = 6$ biological replicates. Statistical analysis: unpaired $t$-test (two-tailed). (Q) Caspase 3/7 activity in DMSO and DHCR7 inhibitor-treated primary human myoblast cells assessed by using a fluorescent reporter. Data were presented as mean ± SEM; $N = 6$ biological replicates. Statistical analysis: unpaired $t$-test (two-tailed). (R) Growth curves of immortal normal human astrocyte cells treated with DHCR7 inhibitor (AY9944) and analyzed by live cell imaging. Data are presented as mean ± SEM; $N = 6$ biological replicates. Statistical analysis: unpaired $t$-test (two-tailed). (S) Caspase 3/7 activity in DMSO and DHCR7 inhibitor-treated immortal normal human astrocyte cells, assessed using a fluorescent reporter. Data are presented as mean ± SEM; $N = 6$ biological replicates. Statistical analysis: unpaired $t$-test (two-tailed). All exact p values are reported in Table EV2. Source data are available online for this figure.

(Fig. 5H), indicating activation of the PERK-eIF2α-ATF4-CHOP axis. Collectively, our findings demonstrate that inhibition of cholesterol biosynthesis in RMS activates UPR pathways and induces the PERK-ATF4-CHOP axis, leading to ER stress-induced intrinsic apoptosis.

## Cholesterol biosynthesis genes are associated with cell cycle progression and poor survival in RMS

For translational validity, we next investigated whether cholesterol biosynthesis genes are clinically relevant in RMS. Analysis of transcriptomic and survival data from 101 RMS patients revealed that high expression of HMGCR and DHCR7 correlated with worse overall survival, as demonstrated by Kaplan–Meier analysis (Fig. 6A,B). To further explore the relevance of these findings in patient-derived tumors, we reanalyzed a recently published single-cell RNA-seq atlas of RMS, which includes both embryonal (FN-RMS) and alveolar (FP-RMS) subtypes derived from primary tumors, patient-derived xenograft (PDX), and in vitro cultured cells (Danielli et al, 2024). DHCR7 and HMGCR were broadly expressed across all malignant cell clusters (Fig. 6C,D), with no specificity for tumor subtype or anatomical origin (Fig. 6E–H). Importantly, gene set enrichment analysis comparing *HMGCR/DHCR7*-positive (cells expressing either or both *HMGCR* and *DHCR7*) versus *HMGCR/DHCR7*-double-negative tumor cells revealed not only enrichment of cholesterol biosynthesis pathways, but also robust enrichment of E2F target genes and G2/M checkpoint regulators (Fig. 6I–K), key pathways we found to be downregulated upon DHCR7 silencing. Thus, the transcriptional programs associated with cholesterol biosynthesis activity in patient-

derived tumors mirror the molecular consequences of DHCR7 inhibition in RMS models, firmly establishing a mechanistic link between cholesterol metabolism, cell cycle progression, and patient outcome, and identifying cholesterol biosynthesis as a clinically relevant therapeutic vulnerability in RMS.

## Discussion

Rhabdomyosarcoma (RMS) remains a formidable pediatric cancer with poor outcomes, particularly in high-risk and relapsed patients. Despite decades of intensive multimodal therapy, RMS survival has plateaued, emphasizing the need for new biologically informed strategies. Our study identifies de novo cholesterol biosynthesis as a central metabolic vulnerability in RMS, revealing its essential role in sustaining proliferation, cell cycle progression, and tumor survival. Silencing DHCR7, the terminal enzyme in cholesterol biosynthesis, profoundly inhibited RMS cell proliferation in vitro, reduced colony formation, and suppressed tumor growth in xenograft models. Mechanistically, DHCR7 depletion triggered cell cycle arrest at the G₂/M phase via the downregulation of E2F transcriptional targets and key proliferation markers, while simultaneously activating ER stress and the PERK–ATF4–CHOP apoptotic axis. These findings establish that cholesterol biosynthesis is not merely a metabolic pathway but a critical driver of proliferative and survival signaling in RMS, directly coupling metabolic flux to core oncogenic processes.

Importantly, RMS cells appear uniquely dependent on de novo cholesterol synthesis, as exogenous LDL supplementation failed to

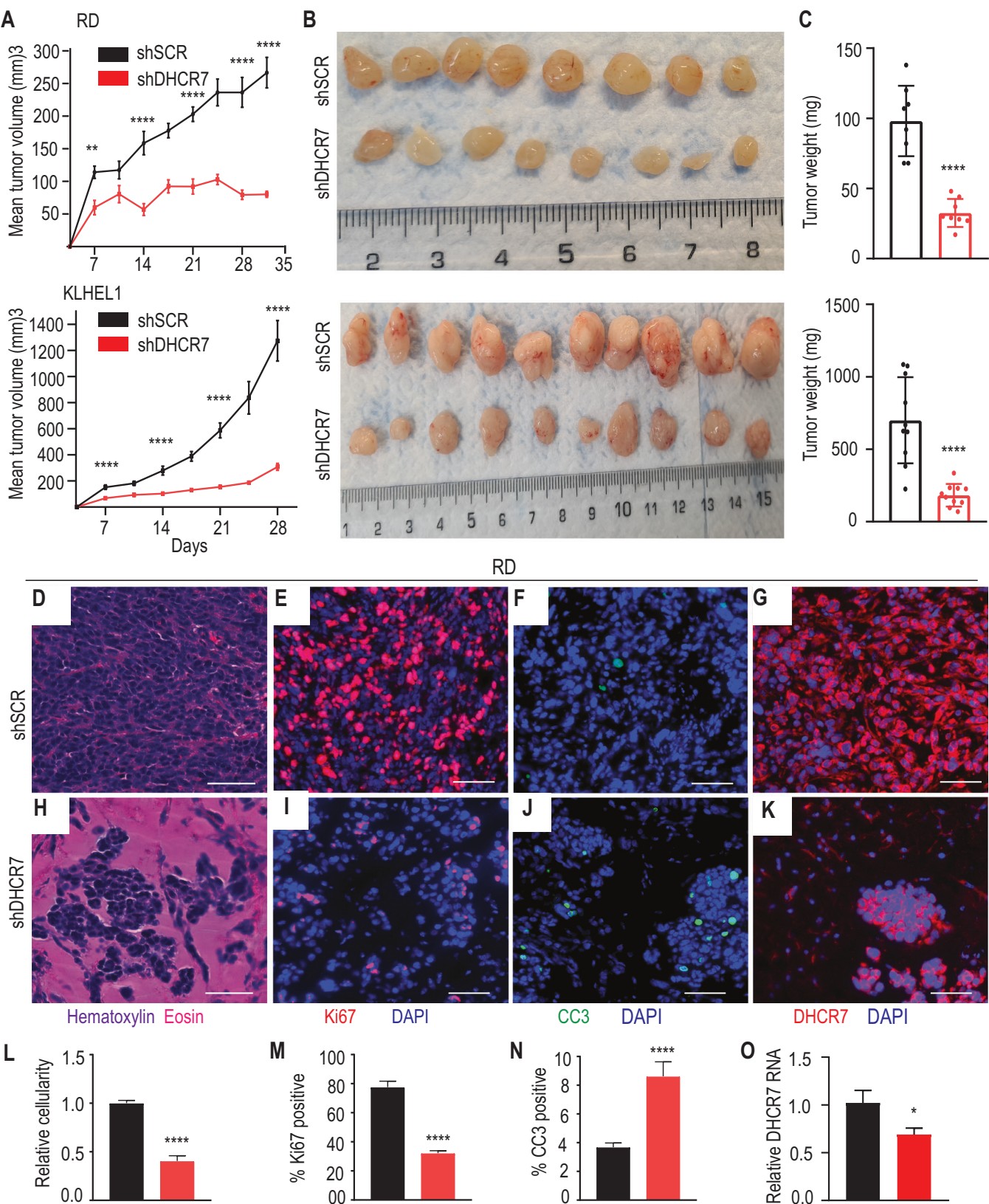

**Figure 3. Cholesterol biosynthesis is essential for RMS tumor xenograft growth.**

(A) Tumor volumes ($mm^3$) in the RD (top) and KLHEL1 (bottom) xenografts, with 8 replicates for RD and 10 for KLHEL1 per group. Statistical analysis: unpaired *t*-test (two-tailed). Significance: **$P < 0.01$, ***$P < 0.001$, ****$P < 0.0001$. (B) Representative images depicting shSCR and shDHCR xenografts derived from RD (Top) and KLHEL1 (Bottom) cells at the end of the experiment. (C) Weights of shSCR and shDHCR7 tumors at the end of the experiment for RD (top) and KLHEL1 (bottom) xenografts. Statistical analysis: unpaired *t*-test (two-tailed). Significance: **$P < 0.01$, ***$P < 0.001$, ****$P < 0.0001$. (D–K) Histological analysis of RD xenografts treated with shSCR (D–G) or shDHCR7 (H–K). Representative H&E-stained sections of the tumors are shown in (D) and (H). Immunofluorescent staining for Ki67 (red) and DAPI (blue) is displayed in (E) and (I), while cleaved caspase 3 (CC3, green) and DAPI (blue) staining is shown in (F) and (J) and DHCR7 (red) (G and K). (L) Quantification of nuclei per tumor area, (M) percentage of Ki67-positive cells, and (N) percentage of cleaved caspase 3 (CC3)-positive cells. (O) mRNA expression of DHCR7 in the excised tumors. Data are presented as mean ± SEM. *$P < 0.05$, and ****$P < 0.0001$. Statistical significance was determined by an unpaired *t*-test (two-tailed). Scale bar = 100 μm. Exact *p* values are reported in Table EV2. Source data are available online for this figure.

rescue growth arrest. This dependency is further reflected by the strong transcriptional upregulation of other cholesterol biosynthesis genes following DHCR7 inhibition, likely mediated by SREBP-driven feedback to low intracellular cholesterol despite the presence of extracellular LDL. In contrast, many other tumor types compensate for reduced endogenous synthesis by upregulating LDL receptor-mediated cholesterol uptake (Duan et al, 2022; Huang et al, 2020), but RMS cells demonstrate minimal metabolic redundancy, underscoring cholesterol biosynthesis as a non-redundant therapeutic target. Consistently, our findings in healthy myoblasts and astrocytes indicate that non-malignant cells predominantly rely on exogenous LDL cholesterol rather than sustain high levels of de novo synthesis. This differential reliance creates a therapeutic window in which pharmacological inhibition of cholesterol biosynthesis could selectively impair RMS cells while sparing normal tissues, thereby minimizing systemic toxicity. Targeting the last enzyme DHCR7 would also make this inhibition even more selective. Together, the tumor-specific dependency of RMS and the compensatory capacity of normal cells strongly position cholesterol metabolism as an exploitable therapeutic vulnerability.

The clinical relevance of the cholesterol biosynthesis pathway is reinforced by analysis of the patient-derived datasets. High expression of HMGCR and DHCR7 correlates with worse overall survival, and single-cell RNA-seq analyses demonstrate broad expression of genes of this pathway across RMS subtypes, irrespective of tumor origin. Importantly, tumor cells expressing HMGCR/DHCR7 were enriched not only for cholesterol biosynthesis signatures but also for E2F target genes and $G_2/M$ checkpoint regulators. This mirrors the molecular consequences of cholesterol biosynthesis inhibition in RMS cells, where we observed suppression of genes involved in both E2F signaling and the $G_2/M$ checkpoint, reflecting disruption of cell cycle progression and linking metabolic dysregulation to tumor aggressiveness. Notably, our results also indicate that PROX1, a transcription factor previously implicated in RMS growth (Gizaw et al, 2022), functions as an upstream regulator of cholesterol biosynthesis, further anchoring this metabolic pathway in the broader transcriptional network that sustains tumor survival.

Our results are in line with, and extend, recent multi-omic analyses of RMS patient samples (Stewart et al, 2018), which highlighted cell cycle checkpoint control and UPR signaling as critical vulnerabilities in RMS. That study demonstrated therapeutic efficacy of pharmacologically targeting the $G_2/M$ checkpoint regulator WEE1, underscoring the importance of mitotic control in RMS survival. By showing that cholesterol biosynthesis inhibition converges on both E2F/$G_2$M checkpoint dysregulation and UPR-driven apoptosis, our work integrates metabolic and signaling perspectives, pointing to a common axis of vulnerability.

Taken together, our results position cholesterol biosynthesis as a high-priority, druggable target in RMS. Future studies should focus on optimizing DHCR7-targeted inhibitors and rationally combining them with cell cycle checkpoint modulators, such as WEE1 inhibitors, or ER stress-enhancing agents to amplify tumor cell killing while minimizing toxicity. Such approaches may overcome resistance to standard therapies and ultimately improve survival in high-risk and relapsed RMS patients. As statins are already widely used and well tolerated, they could be readily added to the existing treatment protocols.

In summary, our work reveals that RMS is uniquely reliant on de novo cholesterol biosynthesis to maintain proliferative signaling and ER homeostasis. Targeting this pathway offers a dual mechanism of action, cell cycle arrest and ER stress–driven apoptosis, highlighting cholesterol metabolism as a promising avenue for novel therapeutic strategies in this aggressive pediatric cancer.

## Methods

**Reagent and tools table**

| Reagent/Resource | Reference or Source | Identifier or Catalog Number |
|---|---|---|
| **Experimental models** | | |
| RD | ATCC | CCL-136 |
| KLHEL1 | Kivelä lab, University of Helsinki | N/A |
| RH30 | Dr. Monika Ehnman, Karolinska Institute | N/A |
| Primary Myoblast | Prof. Heikki Koistinen, University of Helsinki | N/A |
| Immortalized normal human astrocyte | Prof. Pirjo Laakkonen and Dr. Abiodun Ayo | N/A |
| NOD.Cg-*Prkdc^scid Il2rg^tm1Wjl*/SzJ | Jackson Laboratory | #005557 |
| **Recombinant DNA** | | |
| pCMV-VSVg | Addgene | #8454 |
| pCMV-d8.9 | FUGU, University of Helsinki | N/A |
| shHMGCR-1 | FUGU, University of Helsinki | TRCN0000046448 |
| shHMGCR-2 | FUGU, University of Helsinki | TRCN0000046448 |

| Reagent/Resource | Reference or Source | Identifier or Catalog Number |
|---|---|---|
| shDHCR7-1 | FUGU, University of Helsinki | TRCN0000046598 |
| shDHCR7-2 | FUGU, University of Helsinki | TRCN0000046600 |
| shDHCR7-3 | FUGU, University of Helsinki | TRCN0000046602 |
| **Oligonucleotides and other sequence-based reagents** | | |
| PCR primers | This study | Table EV1 |
| **Antibodies** | | |
| Rabbit anti-DHCR7 | Sigma-Aldrich | HPA044280 |
| Mouse anti-HMGCR | Sigma-Aldrich | AMAb90618 |
| Mouse anti-Cyclin D1 | Santa Cruz | sc-8396 |
| Goat anti-PROX1 | R&D Systems | AF2727 |
| mouse anti-Ki67 | Dako | M7240 |
| Rabbit anti-cleaved (active) caspase-3 | R&D | MAB835 |
| Rabbit anti-phospho-eIF2α (Ser51) | Cell Signaling Technology | # 3398 |
| Rabbit anti-eIF2α | Cell Signaling Technology | Cat# 5324 |
| Rabbit anti-PERK | Cell Signaling Technology | Cat# 3192 |
| Rabbit anti-ATF-4 | Cell Signaling Technology | Cat# 11815 |
| Mouse anti-CHOP | Cell Signaling Technology | Cat# 2895 |
| **Chemicals, Enzymes and other reagents** | | |
| AY9944 | MedChemExpress | HY-107420 |
| Lovastatin | MedChemExpress | HY-N0504 |
| IncuCyte Caspase-3/7 reagent | Essen BioScience | #4440 and 4704 |
| Crystal violet | Sigma | #C0775-25G |
| BSA | Biowest | #P6154 |
| DMEM high glucose | Lonza | #BE12614F |
| FBS | Biowest | S181B500 |
| L-glutamine | Corning | #25-005-C1 |
| Matrigel | Corning | #356231 |
| High-Capacity cDNA Reverse Transcription Kit | Thermo Fischer Scientific | #4368813 |
| High pH retrieval solution | DAKO | #S-2367 |
| NucleoSpin RNA II Kit | Macherey-Nagel | #74095 |
| SYBR Green qPCR kit | Roche | #04 913 914 001 |
| SuperSignal West Femto or Pico | Pierce | #34096 #34577 |
| Pierce protease and phosphatase inhibitor | ThermoFisher | #A32961 |
| Pierce®BCA protein assay kit | Thermo Fisher Scientific | #A55860 |
| Pierce protease and phosphatase inhibitor | Thermo Fisher Scientific | #A32961 |
| DMSO | Sigma | #D8418 |
| TNB | PerkinElmer | NEL700001KT |
| Triton-X | Fisher Scientific, | #BP151-500 |

| Reagent/Resource | Reference or Source | Identifier or Catalog Number |
|---|---|---|
| Trypsin-EDTA | Thermo Fisher Scientific | #25200056 |
| Vectashield with Dapi | Vector Laboratories | #H1200 |
| jetPEI | Polyplus | #101000020 |
| Paraformaldehyde (PFA) | Sigma | #PG148 |
| Penicillin and streptomycin | Antibiotics used in cell culture | Lonza, #DE17- 602E |
| **Software** | | |
| ImageLab 6.0.1 software | Bio-Rad | RRID:SCR_014210 |
| GSEA software | http://www.broadinstitute.org/gsea) | RRID:SCR_003199 |
| FlowJo software | https://www.flowjo.com/solutions/flowjo | RRID:SCR_008520 |
| GraphPad Prism (v8.0) | GraphPad | RRID:SCR_002798 |
| Image Studio Lite Version 5 2 Software | LI-COR | RRID:SCR_015795 |
| **Other** | | |

## Human RMS cell lines and constructs

Human RD cells (CCL-136) were acquired from the American Type Culture Collection (ATCC), while KLHEL1 cells were derived from a biopsy of a 27-year-old male diagnosed with high-grade alveolar rhabdomyosarcoma (ARMS) in Kivelä lab. The biopsy was confirmed positive for FKHR (FOXO1) 13q14 gene fusion by fluorescence in situ hybridization (FISH), and the cells expressed vimentin, MYF4, myosin, and desmin. Ethical approval was obtained from the Helsinki University Hospital's ethical committee, and written consent was provided by the patient. RH30 cell lines were kindly provided by Dr. Monika Ehnman (Karolinska Institute, Sweden), and immortalized normal human astrocytes were kindly provided by Prof. Pirjo Laakkonen and Dr. Abiodun Ayo (University of Helsinki). Primary human myoblasts were obtained from Prof. Heikki Koistinen (University of Helsinki). Cells were isolated from vastus lateralis muscle biopsies (~100 mg) of healthy adult male donors under local anesthesia with lidocaine (10 mg/ml, without epinephrine), as described previously (Skrobuk et al, 2012). The human study was approved by the Ethical Committee of the Department of Medicine, Helsinki University Central Hospital, and written informed consent was obtained from all participants. The human experiments conformed to the principles set out in the Declaration of Helsinki and the Department of Health and Human Human Services Belmont Report.

Before use, all cell lines underwent authentication via short tandem repeat (STR) and single nucleotide polymorphism (SNP) profiling using the ForenSeq DNA Signature Kit (Verogen), and regular testing for mycoplasma contamination was conducted. Cells were cultured in Dulbecco's modified Eagle medium (DMEM) high glucose supplemented with 10% fetal bovine serum (FBS), 2 mM L-glutamine, penicillin (100 U/ml), and streptomycin (100 U/ml), and maintained at 37 °C in a humidified 5% $CO_2$ incubator. Lentiviral constructs targeting HMGCR (TRCN0000046448 and TRCN0000046448), DHCR7 (TRCN0000046598, TRCN0000046600, and TRCN0000046602), and a scramble control

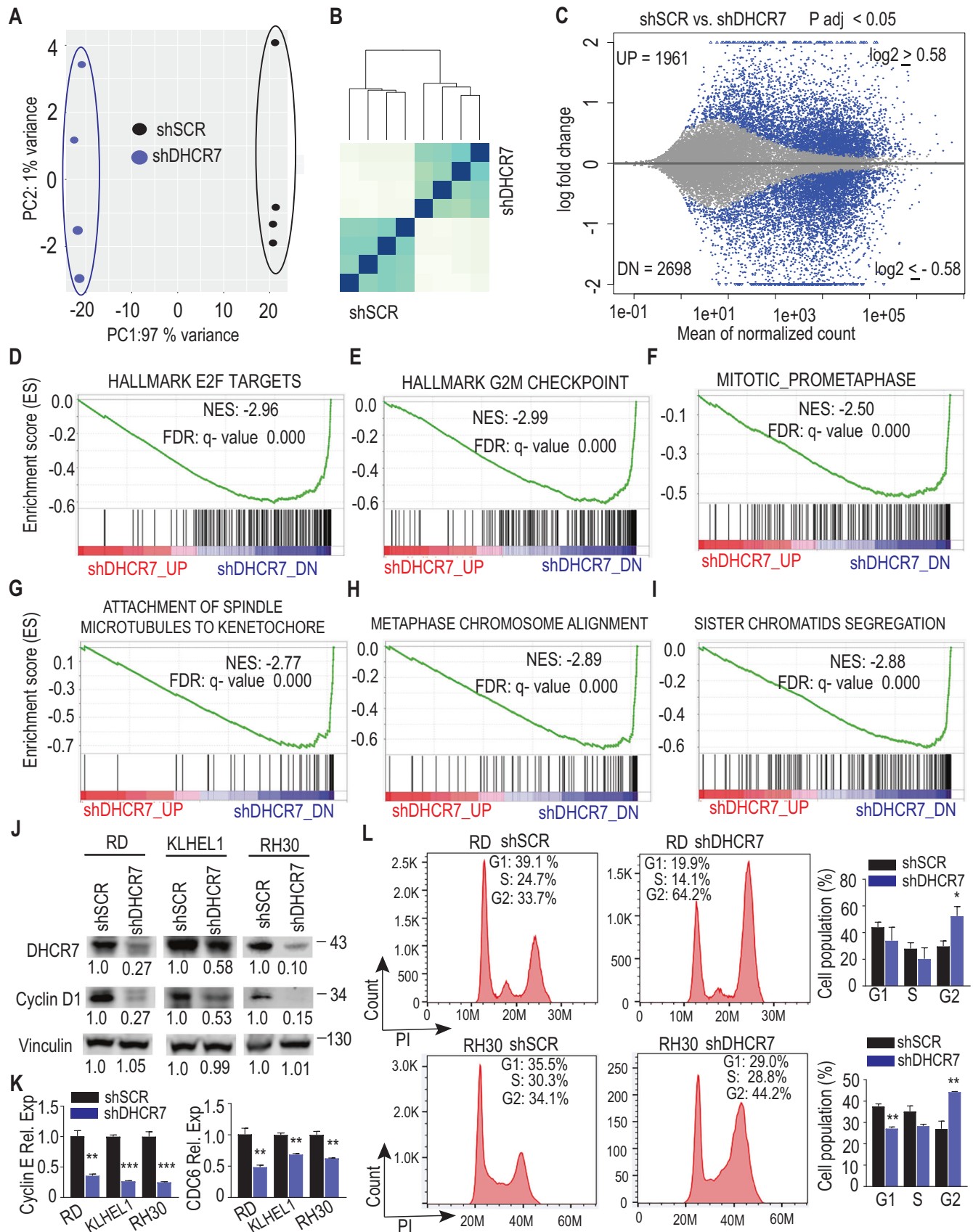

◄ **Figure 4.  Inhibition of cholesterol synthesis impedes cell cycle progression.**

(A) Principal Component Analysis (PCA) illustrating the variance between the DHCR7-silenced RD cells compared to control RD cells. (B) Sample distance analysis of RNA-seq samples ($n = 4 + 4$), depicting the relationship between DHCR7-silenced and control RD cells. (C) MA plot displaying Differentially Expressed Genes (DEGs) in red (FDR < 0.05, log2 fold change (FC) cutoff 0.58) in DHCR7-silenced RD cells versus controls, encompassing 1961 upregulated and 2698 downregulated genes. (D–I) Gene Set Enrichment Analysis (GSEA) plots highlighting the most significantly affected functional categories among downregulated genes in DHCR7-silenced RD cells. The normalized enrichment score (NES) and false discovery rate (FDR) values are shown for each category. (J) Western blot analysis showing that DHCR7 silencing reduces Cyclin D1 protein expression in RD, KLHEL, and RH30 cells. Numbers indicate the relative expression levels of DHCR7, Cyclin D1, and the loading control vinculin in DHCR7-silenced RMS cells. A representative blot from three independent experiments is shown. (K) qPCR analysis of the E2F target cell cycle genes, Cyclin E and CDC6, in DHCR7-silenced versus control RD, KLHEL1, and RH30 cells. Data are presented as mean ± SEM. **$P < 0.01$, ***$P < 0.001$, determined by unpaired two-tailed t-test. (L) Propidium Iodide (PI) flow cytometry analysis of cell cycle distribution, with corresponding statistical analysis, of RD and RH30 cells on the second day post-infection with DHCR7 shRNA. Data are presented as mean ± SEM. *$P < 0.05$, ***$P < 0.001$. Statistical significance was determined using an unpaired two-tailed t-test ($n = 3$ biological replicates per group). Exact $p$ values are reported in Table EV2. Source data are available online for this figure.

(SH00200) were sourced from the TRC library. For viral packaging, 4.5 μg of CMVg, 6.5 μg of CMVd8.9, and 9 μg of shRNA plasmid were transfected into 293FT cells (RRID:CVCL_6911) using jetPEI® transfection reagent. RMS cells were transduced with 293FT supernatant containing 4 μg/ml lentivirus for two to three days. Following transduction, cells were subjected to selection with medium containing 3 μg/ml puromycin for 48 h to establish stable infection.

## RMS cell growth and apoptosis

The growth dynamics and apoptotic responses of HMGCR or DHCR7-silenced cells, alongside their respective controls, were investigated using the IncuCyte live-cell analysis system (Essen Bioscience, Sartorius). Initially, 1000 RD and KLHEL1 cells transduced with lentiviral shHMGCR, shDHCR7, or shSCR were seeded into individual wells of 96-well plates (Corning), with five replicates per construct. Images were acquired every 2 h throughout the experiment.

To explore the impact of HMGCR and DHCR7 chemical inhibition on cell proliferation, 2000–4000 cells were plated in 96-well plates (Corning) with five to six wells per condition. After 24 h, cells were exposed to varying concentrations of Lovastatin (# HY-N0504, MCE) or AY9944 (DHCR7 inhibitor) (# HY-107420, MCE), and images were captured every 2 h to monitor growth dynamics using IncuCyte software. Apoptotic events were monitored by assessing Caspase-3/7 activity. For this, RD and KLHEL1 cells were plated at densities of 4000 to 5000 cells/well in 96-well plates and allowed to adhere for 24 h. Following this, cells were treated with either DMSO or DHCR7 inhibitor, along with IncuCyte Caspase-3/7 Green Apoptosis Assay Reagent (at a 1:1000 dilution). Images were captured every 2 h over a 3-day period to observe apoptotic changes. For cells transduced with shDHCR7 or shSCR, 2000 to 3000 cells were seeded into wells containing media supplemented with IncuCyte Caspase-3/7 Green Apoptosis Assay Reagent (at a 1:1000 dilution), and images were acquired every 2 h for 3 days to assess apoptotic responses.

## Mouse xenograft models

Female NOD scid gamma (NSG) mice (NOD.Cg-Prkdc scid Il2rg tm1Wjl/SzJ, RRID:IMSR_JAX:005557) from Jackson Laboratory were utilized for tumor xenograft experiments and they were randomly assigned to treatment groups. Mice were housed in groups of 3–5 mice at 22 degrees with 12 h light and 12 h dark cycle with ad libitum access to food and water. NSG mice were

anesthetized with isoflurane, and $2 \times 10^6$ shDHCR7 or shSCR RD cells suspended in 100 μl Matrigel were subcutaneously injected into the right and left scapular areas. Similarly, $2 \times 10^6$ shDHCR7 or shSCR KLHEL1 cells in 100 μl Matrigel were injected into the same regions. Mice were regularly monitored for palpable tumor formation, and tumor growth was assessed weekly using calipers to measure height (H), width (W), and depth (D), which were then converted into relative tumor volume. At predefined time points, mice were euthanized, and tumors were excised. The volume and mass of the tumors were measured, and the tumors were subjected to histological and gene expression analysis. All animal procedures were approved by the National Animal Experiment Board in Finland (ESAVI/5365/2022) and performed according to the ARRIVE and local guidelines.

## Gene expression analysis

RNA isolation and cDNA preparation procedures were conducted as described previously (Gizaw et al, 2022). The sequencing data is deposited in Gene Expression Omnibus (RRID:SCR_005012) under accession number GSE279213. Quantitative real-time PCR was performed using a CFX96 Touch Real-Time PCR system. Primer sequences for PCR are provided in Table EV1.

## Histology and immunohistochemistry

Harvested tumors were fixed in 4% paraformaldehyde (PFA), processed, and embedded in paraffin. Sections of 5 μm thickness were then deparaffinized and subjected to staining with either hematoxylin and eosin or specific antibodies. Heat-induced epitope retrieval using High pH Retrieval solution (DAKO) was performed prior to immunostaining for Ki67 (Dako, clone MIB-1, M7240; 1:200), CC3 (R and D Systems Cat# AF835, RRID:AB_2243952, 1:200) and DHCR7 (Sigma-Aldrich Cat# HPA044280, RRID:AB_10794893, 1:250). Signal detection was facilitated by employing Alexa Fluor 488 and 594-conjugated secondary antibodies (Molecular Probes) at a dilution of 1:500. For patient-derived RMS samples, PROX1 (R and D Systems Cat# AF2727, RRID:AB_2170716) staining was carried out utilizing the Bench-Mark XT automated system on 5-μm-thick sections that had undergone antigen retrieval. Imaging was performed using Zeiss Axioplan fluorescent microscope and 3DHISTECH Pannoramic 250 FLASH III digital slide scanner at Genome Biology Unit supported by HiLIFE and the Faculty of Medicine, University of Helsinki, and Biocenter Finland. Image processing and analysis

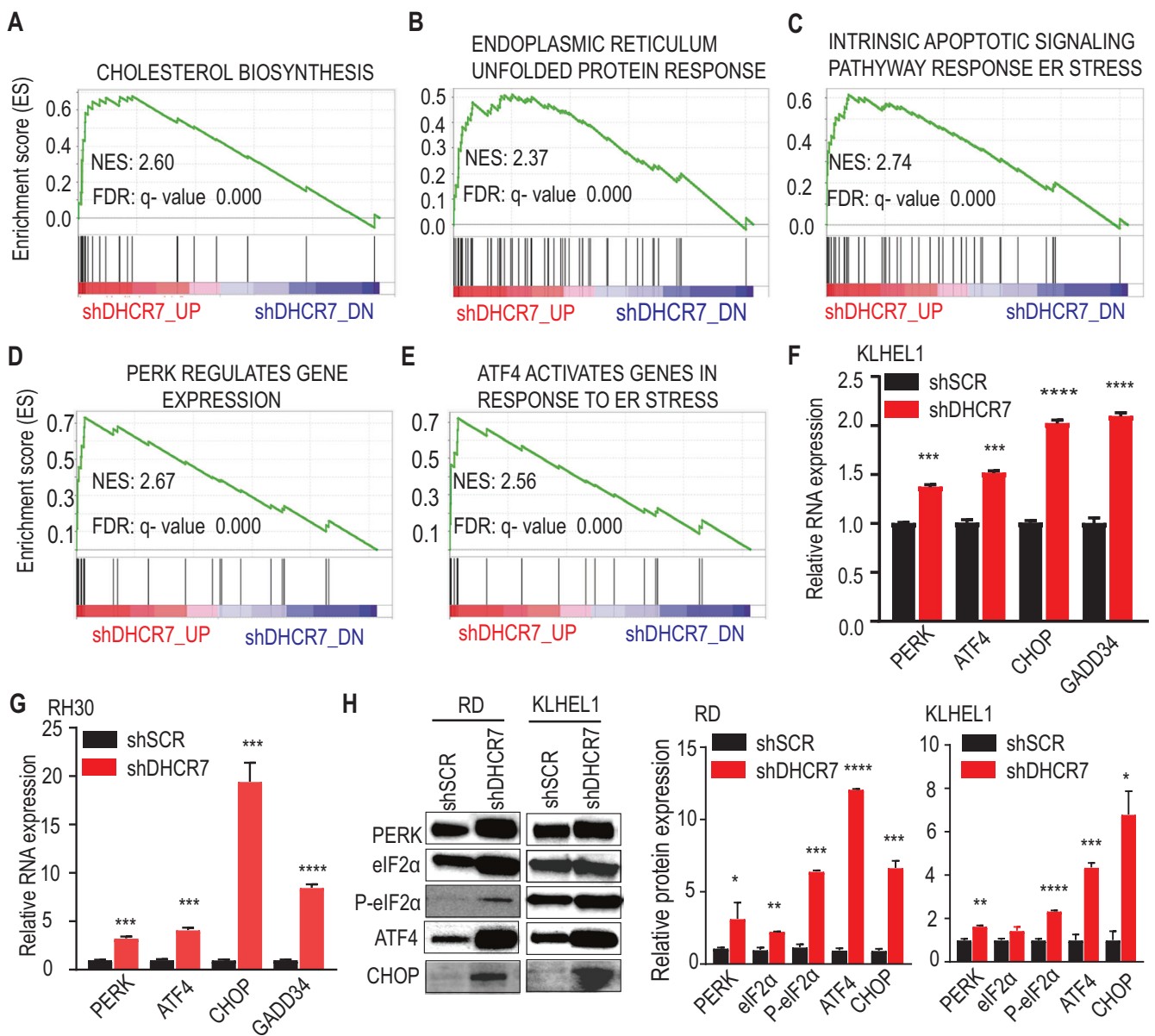

**Figure 5. Inhibition of cholesterol biosynthesis induces ER stress-mediated apoptosis.**

(A–E) Gene Set Enrichment Analysis (GSEA) plots showing the most significantly enriched functional categories among upregulated genes in DHCR7-silenced RD cells. Normalized enrichment score (NES) and False Discovery Rate (FDR) are indicated for each category. (F, G) Real-time qPCR analysis of gene expression levels involved in the ER stress-induced apoptotic pathway following DHCR7 knockdown in KLHEL1 cells (F) and RH30 cells (G). Data were presented as mean ± SEM; $N = 3$ biological replicates. Statistical analysis: unpaired $t$-test (two-tailed). Significance: ***$P < 0.001$, ****$P < 0.0001$. (H) Western blot analysis showing phosphorylation of eIF2α and expression of proteins associated with the ER stress-induced apoptotic pathway in RD and KLHEL1 cells after DHCR7 knockdown. Data are presented as mean ± SEM; $N = 3$ biological replicates. Statistical analysis: unpaired $t$-test (two-tailed). Significance is indicated: *$P < 0.05$, **$P < 0.01$, ***$P < 0.001$, ****$P < 0.0001$. Exact $p$ values are reported in Table EV2. Source data are available online for this figure.

were carried out using ImageJ software (RRID:SCR_002285). Investigators performing the imaging and analysis were blinded to group allocation to prevent observer bias.

**Immunoblots**

Total protein from human RMS cells was extracted using RIPA buffer (50 mM Tris-HCl pH 7.6, 150 mM NaCl, 1% NP-40, 0.5% sodium deoxycholate, 0.1% SDS) supplemented with protease and phosphatase inhibitors (Pierce). Protein concentrations were determined using the BCA Protein Assay Kit (Pierce Biotechnology, Rockford, IL, USA). Equal amounts of protein (8–10 μg) were denatured at 95 °C, resolved on 4–20% TGX gels (Bio-Rad; Criterion TGX Stain-Free Cat# 5678094 or Mini-PROTEAN TGX) at 270 V for 35 min, and transferred to low-fluorescence PVDF or nitrocellulose membranes. Membranes were incubated overnight at 4 °C with primary antibodies against phospho-

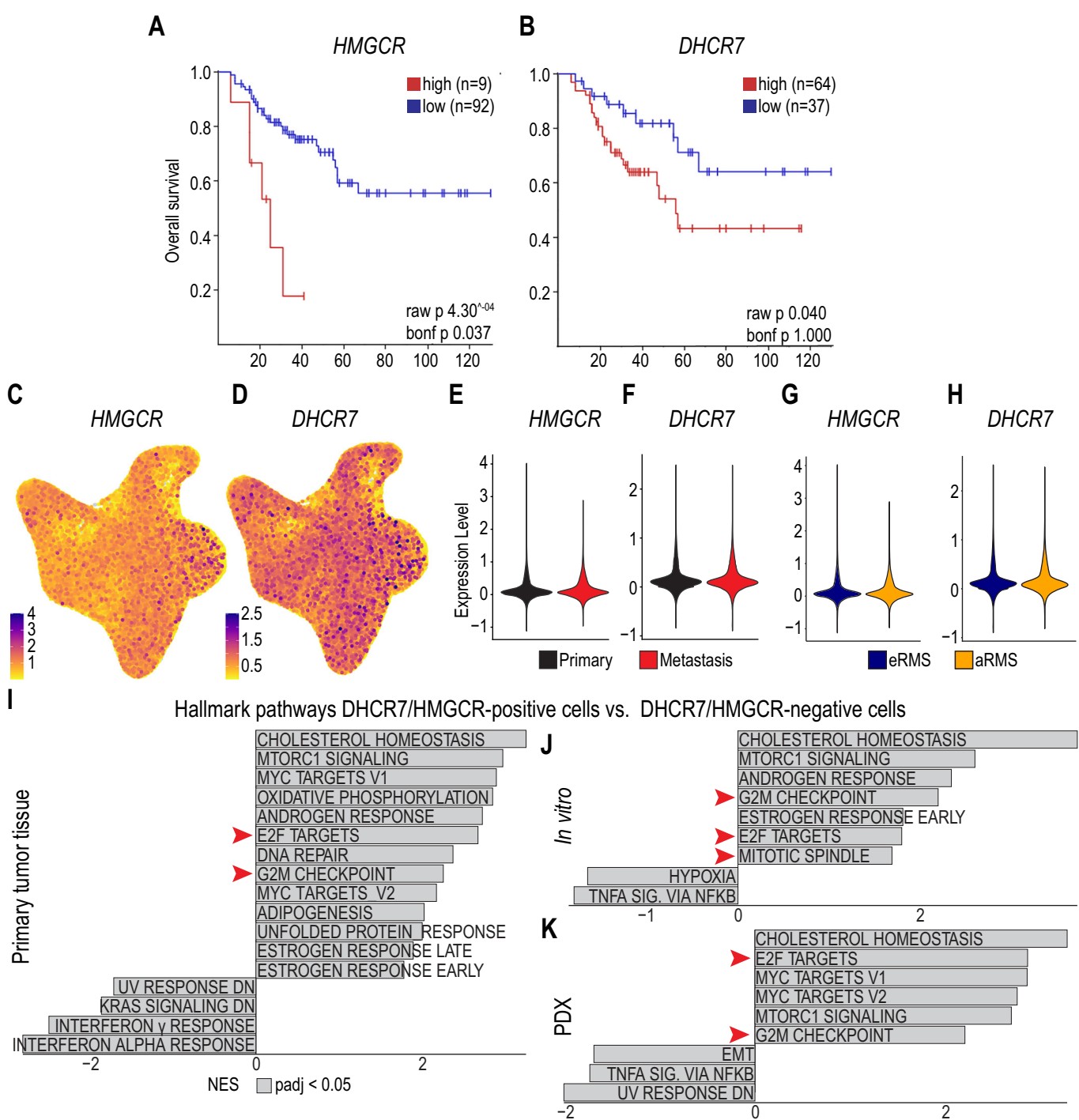

**Figure 6. High *HMGCR* or *DHCR7* expression correlates with poor survival.**

(**A, B**) RMS 5-year overall survival was analyzed in patients with high or low expression of *HMGCR* (**A**) or *DHCR7* (**B**). (**C–K**) RMS scRNA-seq atlas was reanalyzed. (**C, D**) Feature plots showing the expression of *HMGCR* (**C**) and *DHCR7* (**D**) in merged RMS patient samples. (**E–H**) Violin plots showing the expression of *HMGCR* (**E, G**) and *DHCR7* (**F, H**) in pooled samples of primary tumors and in metastatic lesions (**E, F**) and in eRMS or aRMS (**G, H**). (**I–K**) Differentially expressed genes between *HMGCR;DHCR7*-positive and *HMGCR;DHCR7*-double-negative cells in primary tumor samples (**I**), in vitro tumor cell samples (**J**), or in PDX samples (**K**), analyzed by using GSEA. Shown are normalized enrichment scores (NES) with padj. <0.05.

eIF2α (Ser51) (Cell Signaling Technology Cat# 3398, RRID:AB_2096481, 1:1000), total eIF2α (Cell Signaling Technology Cat# 5324, RRID:AB_10692650, 1:1000), PERK (Cell Signaling Technology Cat# 3192, RRID:AB_2095847, 1:1000), ATF4 (Cell Signaling Technology Cat# 11815, RRID:AB_2616025, 1:1000), CHOP (Cell Signaling Technology Cat# 2895, RRID:AB_2089254, 1:1000), DHCR7 (Sigma-Aldrich Cat# HPA044280, RRID:AB_10794893, 1:1000), HMGCR (Sigma-Aldrich Cat# AMAb90618, 1:1000), Cyclin D1 (Santa Cruz Cat# sc-8396, RRID:AB_627327, 1:500), and Vinculin (Sigma-Aldrich Cat# V9131, RRID:AB_477629, 1:2500). HRP-conjugated secondary antibodies were applied for 1 h at room temperature (Jackson ImmunoResearch, 1:10,000, RRID:AB_2337891; Dako, 1:5000, RRID:AB_579779). Protein signals were visualized using SuperSignal West Femto Maximum Sensitivity Substrate (Pierce Biotechnology) and imaged with a ChemiDoc MP Imaging System (Bio-Rad, RRID:SCR_019037). Protein normalization was performed using either vinculin as a loading control or stain-free total protein quantification with ImageLab 6.0.1 software (Bio-Rad, RRID:SCR_014210).

## RNA sequencing and gene set enrichment analysis

RNA-seq and Gene Set Enrichment Analysis (GSEA) were conducted for shPROX1 KLHEL1 cells and human myoblasts, along with their respective controls, as described previously (Gizaw et al, 2022). For shDHCR7 and control RNA-seq, RNA samples were obtained from RD cells, and messenger RNA (mRNA) was purified from total RNA using poly-T oligo-attached magnetic beads. Subsequently, the mRNA was fragmented, and the first strand cDNA was synthesized using random hexamer primers, followed by the synthesis of the second strand cDNA. The library preparation involved end repair, A-tailing, adapter ligation, size selection, amplification, and purification steps. The quality and quantity of the library were assessed using Qubit and real-time PCR for quantification, as well as a bioanalyzer for size distribution detection. The quantified libraries were pooled and sequenced on Illumina platforms based on effective library concentration and data amount criteria. The analysis of the generated FASTQ data was performed using Chipster software (www.chipster.csc.fi) (Kallio et al, 2011) as previously described. Gene Set Enrichment Analysis (GSEA) (Subramanian et al, 2005) was also conducted using the GSEA software (RRID:SCR_003199) (http://www.broadinstitute.org/gsea), as described previously.

## Cell cycle assay

For cell cycle analysis, RD cells were harvested and washed with cold PBS. The cells were then fixed in 70% cold ethanol and stored at −20 °C overnight. Following fixation, the cells were stained with propidium iodide and subjected to flow cytometry using the NovoCyte Quanteon 4025. Flow cytometry data were analyzed using FlowJo software (RRID:SCR_008520).

## In silico data analysis

Publicly available dataset on RMS tumors deposited in Gene Expression Omnibus (https://www.ncbi.nlm.nih.gov/geo/, GSE108022) as used to analyze the expression of mevalonate pathway enzymes in healthy skeletal muscle and in FN- and FP-RMS tumors, Kaplan–Meier survival analyses were performed using "Tumor Rhabdomyosarcoma - Williamson - 101 - MAS5.0 -

### The paper explained

**Problem**

Rhabdomyosarcoma (RMS) is the most common soft tissue sarcoma of childhood, yet outcomes remain poor in high-risk and relapsed patients despite intensive multimodal therapy. Current treatments do not exploit tumor-specific metabolic dependencies, and the molecular drivers of therapy resistance remain incompletely understood. Identifying selective vulnerabilities that distinguish RMS from normal tissues is therefore critical for developing more effective and less toxic therapeutic strategies.

**Results**

Our study uncovers de novo cholesterol biosynthesis as an essential metabolic dependency in RMS. Silencing DHCR7, the terminal enzyme in this pathway, profoundly impaired RMS growth both in vitro and in xenograft models by inducing $G_2$/M cell cycle arrest through down-regulation of E2F targets, while simultaneously activating ER stress-mediated apoptosis. Importantly, this dependency was not rescued by exogenous LDL cholesterol, highlighting a non-redundant reliance on cholesterol biosynthesis. Clinical and single-cell transcriptomic data confirmed that high HMGCR and DHCR7 expression correlates with poor survival and enrichment of cell cycle checkpoint signatures, mirroring our experimental findings.

**Impact**

These results establish cholesterol biosynthesis as a tumor-specific vulnerability in RMS, mechanistically linking metabolic reprogramming to cell cycle dysregulation and survival signaling. Unlike normal myoblasts and astrocytes, which rely primarily on exogenous cholesterol uptake, RMS cells exhibit unique dependence on de novo cholesterol synthesis, suggesting a therapeutic window. By integrating metabolic and signaling perspectives, our findings nominate cholesterol biosynthesis inhibitors—alone or in rational combination with cell cycle checkpoint modulators or ER stress-enhancing agents—as promising therapeutic strategies to improve outcomes in this aggressive pediatric cancer.

u133p2" dataset containing data from 101 patients in the R2 platform (https://hgserver1.amc.nl/).

## Single-cell RNA-sequencing analysis

For analysis of transcriptional landscape of RMS patient samples, we reanalyzed published RMS single-cell RNA-sequencing (scRNA-seq) atlas (Danielli et al, 2024). For analysis presented in this study, we used cell cluster annotations made by the atlas authors. To analyze which gene signatures were enriched or decreased, we performed GSEA by using MSigDB Hallmark gene signatures. Module enrichment scores were quantified using Seurat's AddModuleScore function.

## Statistics

All cell culture experiments were conducted at least 2–4 times (with the same cells and constructs or using another construct for the same gene or another cell line). Excluding RNA sequencing, statistical analyses were conducted using GraphPad Prism (v8.0) (RRID:SCR_002798). Normality was tested using Shapiro–Wilk test and for comparisons between two groups, and an unpaired two-tailed t-test was applied. For experiments with more

than two groups, one-way ANOVA followed by Tukey's or Dunnett's multiple comparisons test was performed. Values are expressed as mean ± SE. Statistical significance levels were set at *$P < 0.05$; **$P < 0.01$; ***$P < 0.001$ and ****$P < 0.0001$. All used statistical tests for each experiment and exact *P*-values are reported in the Table EV2.

## Graphics

Synopsis graphics was created with BioRender.com.

# Data availability

The datasets produced in this study are available in the following databases: RNA sequencing datasets: Gene Expression Omnibus (GEO) under accession number GSE279213.

The source data of this paper are collected in the following database record: biostudies:S-SCDT-10_1038-S44321-025-00336-x.

# Peer review information

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

# Acknowledgements

We would like to thank Mari Jokinen, Maria Arrano, Tanja Laakkonen, and Tapio Tainola for their excellent technical help. Elina Ikonen is acknowledged

for discussions and comments during the project and for DHCR7 antibody testing. We also thank the Laboratory Animal Center at the University of Helsinki for expert animal care, the Biomedicum Imaging Unit for microscope support and the Biomedicum Virus Core and Genome Biology Unit for TRC library construct and virus preparation. The work was funded by the Cancer Foundation Finland sr, Barncancerfonden, Children's Cancer Foundation Väre, Children's Cancer Foundation Aamu, K. Albin Johanssons stiftelse sr., and Orion Research Foundation sr. Open access funded by Helsinki University Library.

## Author contributions

**Nebeyu Yosef Gizaw**: Conceptualization; Data curation; Formal analysis; Funding acquisition; Investigation; Visualization; Methodology; Writing—original draft; Writing—review and editing. **Kalle Kolari**: Data curation; Formal analysis; Investigation; Methodology. **Pauliina Kallio**: Data curation; Formal analysis; Investigation; Visualization; Methodology; Writing—review and editing. **Kari Alitalo**: Resources; Supervision; Methodology; Writing—review and editing. **Riikka Kivelä**: Conceptualization; Resources; Data curation; Supervision; Funding acquisition; Writing—original draft; Project administration; Writing—review and editing.

Source data underlying figure panels in this paper may have individual authorship assigned. Where available, figure panel/source data authorship is listed in the following database record: biostudies:S-SCDT-10_1038-S44321-025-00336-x.

## Disclosure and competing interests statement

The authors declare no competing interests.

# Expanded View Figures

**Figure EV1.   Inhibition of cholesterol biosynthesis impairs rhabdomyosarcoma (RMS) cell growth.**

(**A**) Quantitative PCR analysis showing HMGCR mRNA expression levels in RD cells transduced with shSCR (control) or two independent shHMGCR shRNA constructs. Data were presented as mean ± SEM; $N = 3$ biological replicates. Statistical analysis: one-way ANOVA analysis followed by Dunnett's multiple comparison test. Significance: **$P < 0.01$, ***$P < 0.001$. (**B**) Cell growth was monitored via IncuCyte live cell imaging for RD cells stably expressing shSCR or shHMGCR across two silencing constructs. Data were presented as mean ± SEM; $N = 3$ biological replicates. Statistical analysis: one-way ANOVA analysis followed by Dunnett's multiple comparison test. Significance: ***$P < 0.001$, ****$P < 0.0001$. (**C, D**) Colony formation assay in RD cells, along with its quantification. Data were presented as mean ± SEM; $N = 3$ biological replicates. Statistical analysis: one-way ANOVA analysis followed by Dunnett's multiple comparison test. Significance: ****$P < 0.0001$. (**E**) qPCR analysis of HMGCR mRNA levels in KLHEL1 cells following transduction with shSCR (control) or two independent shHMGCR constructs. Data were presented as mean ± SEM; $N = 3$ biological replicates. Statistical analysis: one-way ANOVA analysis followed by Dunnett's multiple comparison test. Significance: **$P < 0.001$, ****$P < 0.0001$. (**F**) Cell growth in KLHEL1 cells expressing shSCR or shHMGCR was tracked using IncuCyte live imaging. Data were presented as mean ± SEM; $N = 3$ biological replicates. Statistical analysis: one-way ANOVA analysis followed by Dunnett's multiple comparison test. Significance: **$P < 0.001$, ****$P < 0.0001$ (**G, H**) Colony formation assay and its corresponding quantification in KLHEL1 cells. Data were presented as mean ± SEM; $N = 3$ biological replicates. Statistical analysis: one-way ANOVA analysis followed by Dunnett's multiple comparison test. Significance: **$P < 0.001$, ****$P < 0.0001$. (**I, J**) Growth curves of RD (**I**) and KLHEL1 (**J**) cells treated with different concentrations of lovastatin, analyzed through live cell imaging. Data were presented as mean ± SEM; $N = 3$ biological replicates. Statistical analysis: one-way ANOVA analysis followed by Dunnett's multiple comparison test. Significance: ****$p < 0.0001$. Exact $p$ values are reported in Table EV2.

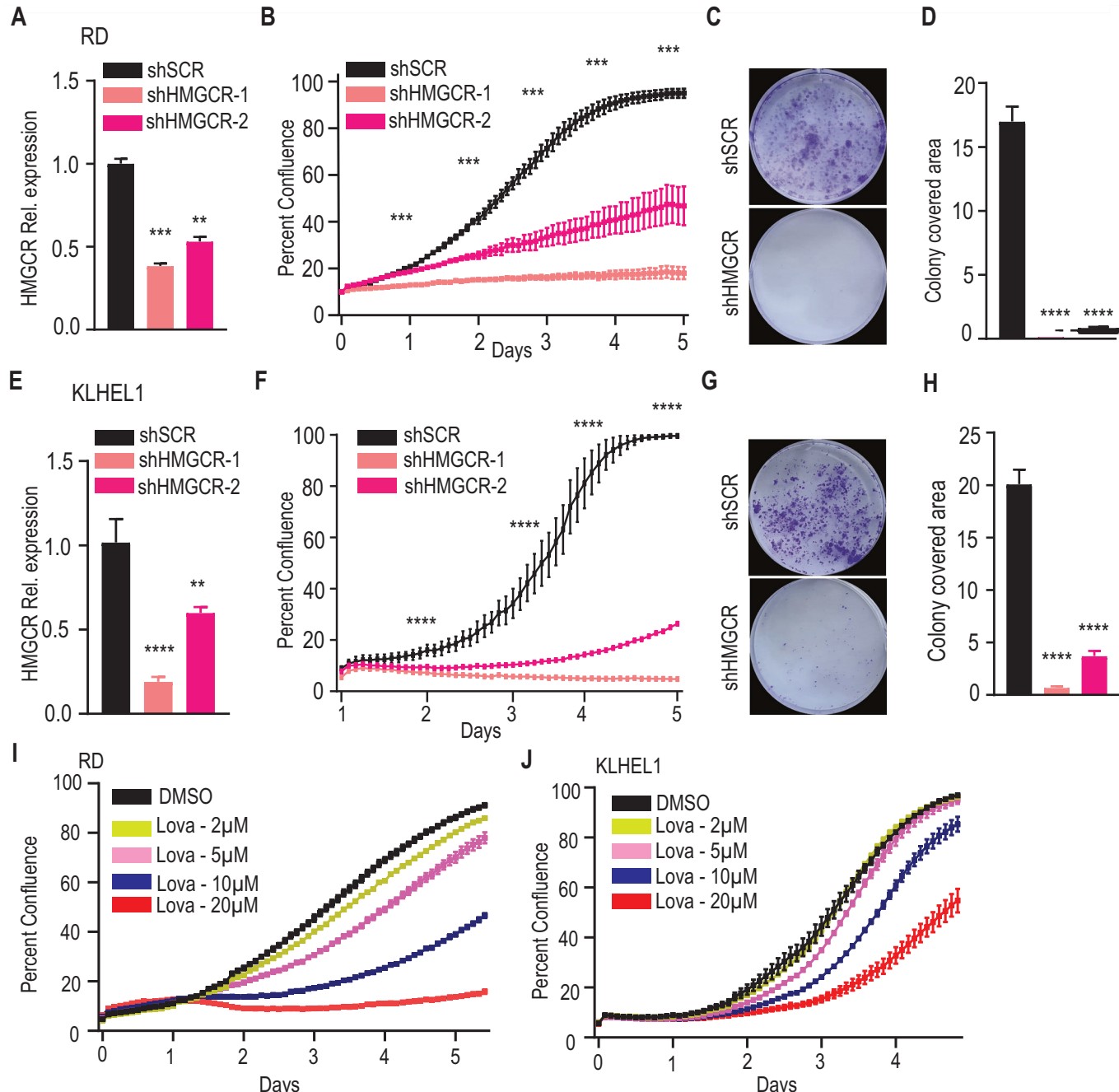

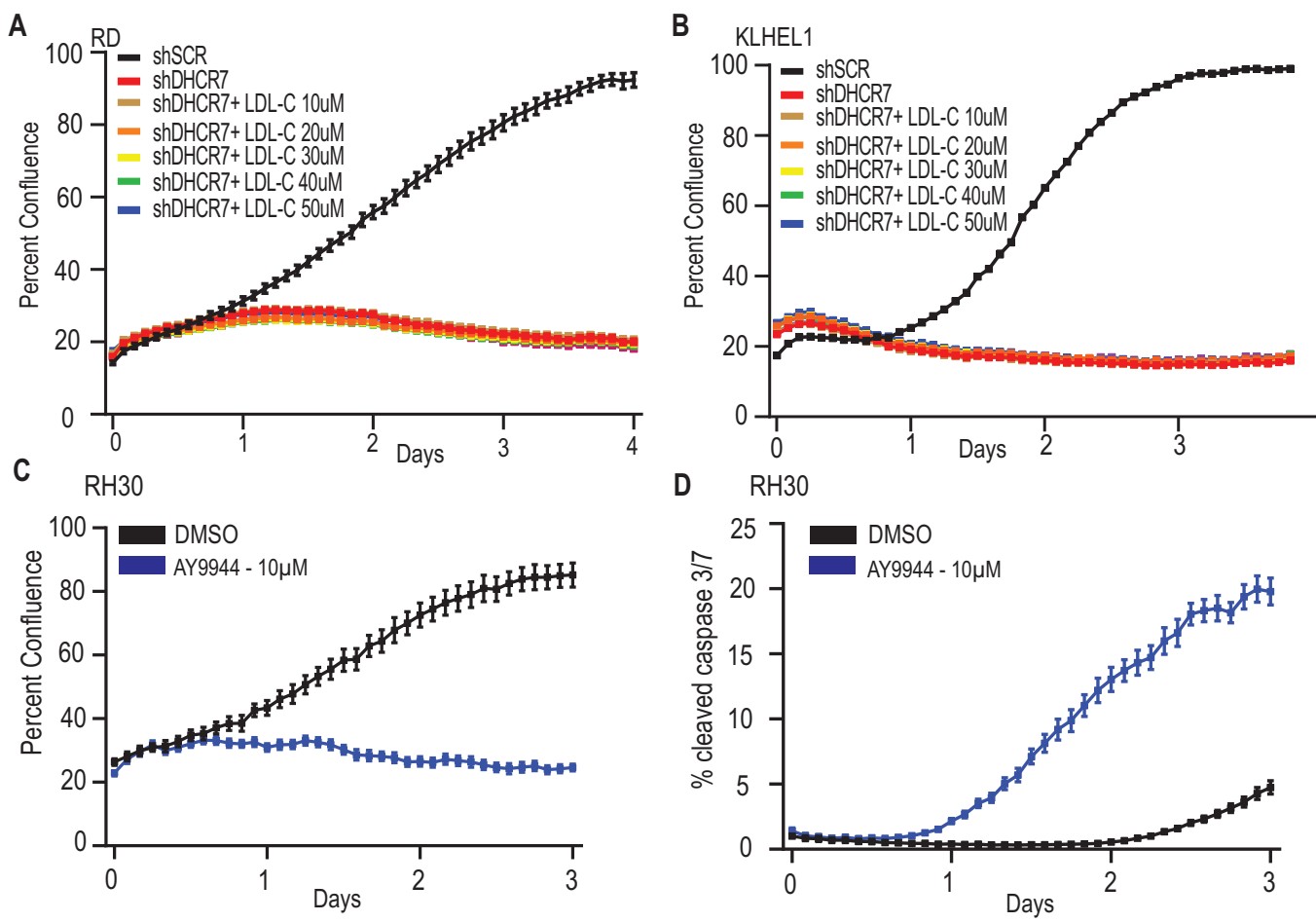

**Figure EV2. De novo cholesterol biosynthesis is essential for RMS growth and survival.**

(A, B) Supplementation with varying concentrations of LDL cholesterol (LDL-c) could not reverse the antiproliferative effects of cholesterol biosynthesis inhibition with shDHCR7 in RD (A) and KLHEL1 (B) cells. (C) Growth curves of RH30 cells treated with different concentrations of DHCR7 inhibitor (AY9944) and analyzed by live cell imaging. (D) Caspase 3/7 activity in dimethyl sulfoxide (DMSO) and DHCR7 inhibitor-treated RH30 cells. Data are presented as mean ± SEM. ****$P < 0.0001$. Statistical significance was determined using an unpaired two-tailed t-test ($n = 3$ per group). Exact $p$ values are reported in Table EV2.

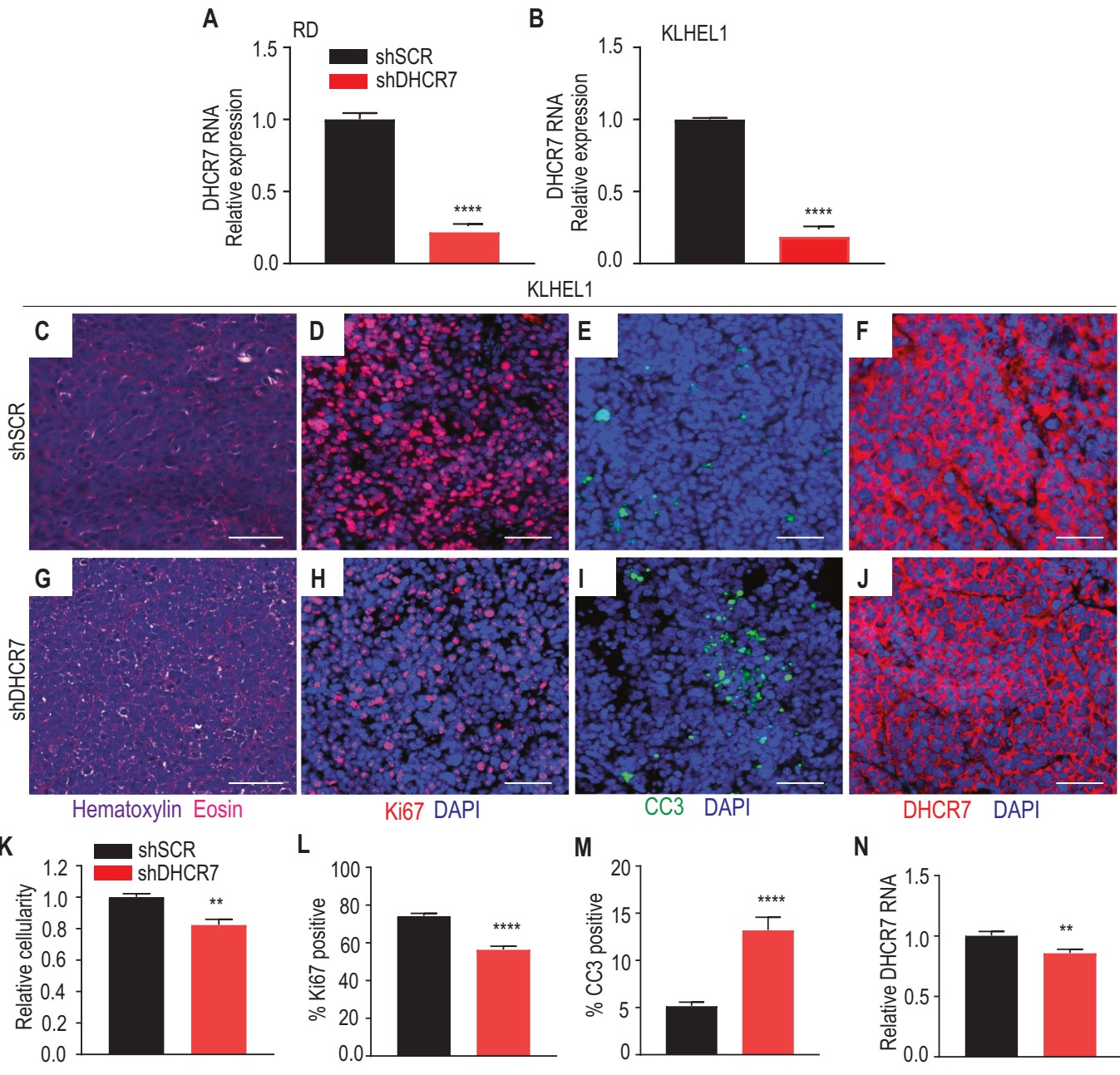

**Figure EV3.   Cholesterol biosynthesis is essential for RMS tumor xenograft growth.**

(**A, B**) qPCR analysis of DHCR7 mRNA expression in shDHCR7 and shSCR-transduced KLHEL1 and RD cells prior to tumor implantation. Data were presented as mean ± SEM; $N = 3$ biological replicates. Statistical analysis: unpaired $t$-test (two-tailed). Significance: ****$P < 0.0001$. (**C, J**) Histological analysis of KLHEL1 xenograft tumors derived from cells transduced with shSCR (**A–D**) or shDHCR7 (**G–J**). Representative images of H&E-stained tumor sections (**C, G**), Ki-67 immunostaining (red) for proliferation and DAPI (blue) (**D, H**), cleaved caspase 3 (CC3) immunostaining (green) and DAPI (blue) for apoptosis detection (**E, I**) and DHCR7 (red) and DAPI (blue) (**F, J**). (**K**) Quantification of nuclei count per tumor area. Data were presented as mean ± SEM; $N = 10$ mice. Statistical analysis: unpaired $t$-test (two-tailed). Significance: **$P < 0.01$. (**L**) Percentage of Ki67-positive proliferating cells within the tumors. Data were presented as mean ± SEM; $N = 10$ mice. Statistical analysis: unpaired $t$-test (two-tailed). Significance: ****$P < 0.0001$. (**M**) Percentage of activated caspase 3 (CC3)-positive cells, indicating apoptosis. Data were presented as mean ± SEM; $N = 10$ mice. Statistical analysis: unpaired $t$-test (two-tailed). Significance: ****$P < 0.0001$. (**N**) qPCR analysis of DHCR7 mRNA expression in tumor tissue. Data were presented as mean ± SEM; $N = 10$ mice. Statistical analysis: unpaired $t$-test (two-tailed). Significance: **$P < 0.01$. Exact $p$ values are reported in Table EV2. Scale bar: 100 μm.

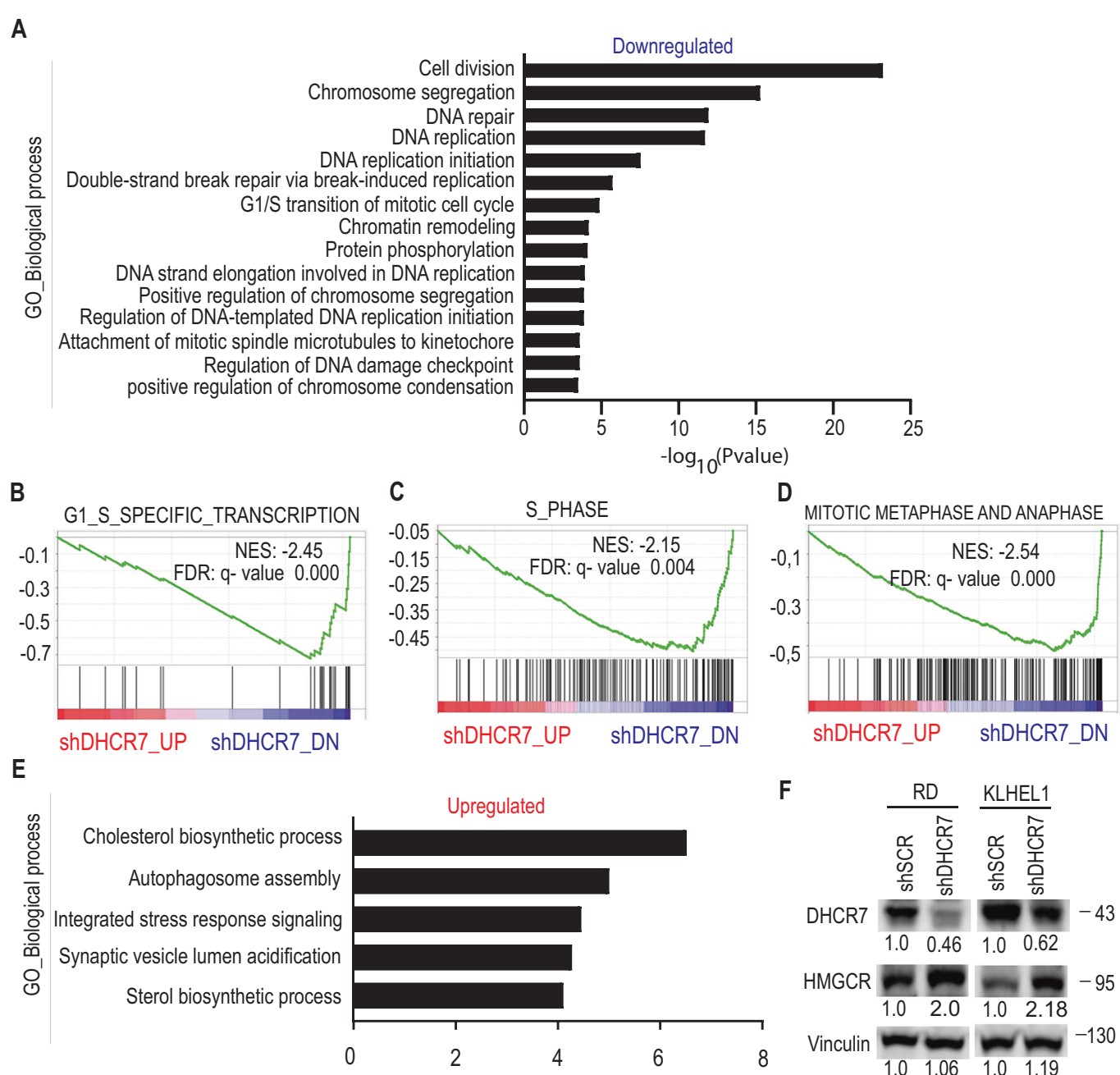

**Figure EV4. Cholesterol biosynthesis inhibition disrupts cell cycle progression and induces an integrated stress response.**

(A) Gene Ontology (GO) analysis showing the most significantly enriched functional categories among the downregulated genes in DHCR7-silenced cells. Statistical analysis: modified Fisher's test. (B–D) Gene Set Enrichment Analysis (GSEA) plots illustrating the top functional categories significantly affected in DHCR7-silenced RD cells. Normalized enrichment score (NES) and False Discovery Rate (FDR) values are shown for each pathway. (E) GO analysis showing the most significantly enriched functional categories among the upregulated genes in DHCR7-silenced cells. (F) Western blot analysis demonstrating that DHCR7 silencing induces compensatory upregulation of HMGCR in RD and KLHEL1 cells, reflecting their reliance on de novo cholesterol biosynthesis. Numbers indicate relative expression levels of DHCR7 and HMGCR normalized to Vinculin. Representative blots from three independent experiments are shown.

