## [Peer Review File · EMBO Molecular Medicine]

Inhibiting cholesterol synthesis halts rhabdomyosarcoma growth via ER stress and cell cycle arrest

Nebeyu Gizaw, Kalle Kolari, Pauliina Kallio, Kari Alitalo, and Riikka Kivelä

Corresponding author: Riikka Kivelä (riikka.m.kivela@jyu.fi)

Review Timeline:

Transferred from Review Commons:	28th Apr 25
Editorial Decision:	29th Apr 25
Revision Received:	8th Sep 25
Editorial Decision:	30th Sep 25
Revision Received:	25th Oct 25
Accepted:	27th Oct 25

Editor: Zeljko Durdevic

Transaction Report:

Review
COMMONS

This manuscript was transferred to EMBO Molecular Medicine following peer review at Review Commons.

Review #1**1. Evidence, reproducibility and clarity:****Evidence, reproducibility and clarity (Required)**

The manuscript by Gizaw et al characterizes the cholesterol biosynthetic pathway and the effect of its knockdown or inhibition on rhabdomyosarcoma tumor properties. The Authors find that the PROX1 transcription factor mediated cholesterol biosynthesis regulates rhabdomyosarcoma cell growth and proliferation. Blocking the cholesterol biosynthetic pathway leads to reduced proliferation, cell cycle arrest and ER-stress mediated enhanced apoptosis. Detailed transcriptomic analysis indicate gene expression patterns that support these findings.

****Major comments****

1. Details of the healthy human myoblasts that are used in Figure 1A are not provided and should be updated. Evidence of PROX1 knockdown should be presented. What kind of pathways and gene ontology predictions were associated with the 225 genes that are commonly downregulated between all three cell lines in Figure 1A?
2. In Figure 2, while the effect of the shRNAs targeting DHRC7 or the DHRC7 inhibitor AY9944 are striking, it is not clear whether these effects are specific to rhabdomyosarcoma cells or cancer cells. A control, human myoblast cell line or another non-cancerous cell line should be used to repeat these experiments quantifying Caspase3/7 activity, cell growth etc. to assess the cancer cell specificity of such treatments. Evidence of DHRC7 knockdown at the protein level would add to the study.
3. Western blots for Caspase3 quantification and a cell proliferation marker such as Cyclin D in shSCR and shDHRC7 tumor lysates would validate the data shown in the Figure 3. Are the shRNA constructs used inducible ones? If not, how do the Authors distinguish the effect of shDHRC7 on tumor engraftment versus tumor proliferation and growth? Many of the graphs need proper labeling of the axes and what the bars represent.
4. Gene ontology and pathway analysis will add to Figure 4.
5. In Figure 5A, how do the Authors explain the upregulation of cholesterol biosynthetic pathway genes upon shDHRC7 treatment? Are these effects seen at the protein level and if alternate pathways maintain cholesterol biosynthesis, how do the Authors think this strategy will be viable to treat such tumors? In Figure 5G-H, was a loading control used? If so, blots for that should be included.
6. Lines 286-287 refers to Figure S1G, H; it should be corrected to Figure S1I, J.

2. Significance:

Significance (Required)

Based on my expertise on rhabdomyosarcoma tumors, the manuscript is clear, concise and provides a significant advance to the field. Detailed mechanistic characterization is lacking, which takes away some of the significance of the findings, but the work done stands alone as description of the effect of the cholesterol biosynthetic pathway in rhabdomyosarcoma. Another aspect to be considered by the Authors is the potential specificity of targeting a ubiquitous pathway such as cholesterol biosynthesis, which is important to most cells and not only cancer cells. Overall, the manuscript may be revised to address the specific comments below.

3. How much time do you estimate the authors will need to complete the suggested revisions:

Estimated time to Complete Revisions (Required)

(Decision Recommendation)

Between 1 and 3 months

Yes

Review #2

1. Evidence, reproducibility and clarity:

Evidence, reproducibility and clarity (Required)

In this manuscript entitled "Targeting de novo cholesterol synthesis in rhabdomyosarcoma induces cell cycle arrest and triggers apoptosis through ER stress-mediated pathways" Gizaw et al investigate the crucial effect of targeting cholesterol biosynthesis in RMS. While this manuscript gives novel insights into putative therapeutic approach, there are some comments that should be address by the authors.

****Major comments****

1. The author demonstrated a correlation between PROX1 levels and the cholesterol synthesis pathway. Which genes from the pathway are mostly affected? The manuscript could benefit from a graphical representation of the pathway showing up- and downregulated genes from the RNAseq analysis. This will help in understanding why the authors decided to study HMGCR silencing as shown in Supplementary Figure 1A.
2. Based on the previous comment, are the genes from the cholesterol synthesis identified in the RNA-seq similar to those detected in the publicly available data set presented in Figure 1E? In addition, validation of changes of these genes should be performed in the RMS cell lines as well as in myoblasts.
3. In Figure 3, the authors study the impact of DHCR7-silencing in tumor growth in vivo. Please, provide stainings also for DHCR7 to show that cells indeed have silenced DHCR7.
4. In Figure 4, the RNAseq data revealed downregulation in E2F genes as well as genes involved in cell cycle progression. It would be important that the authors provide examples of these genes and validate this data by performing qPCR.
5. In Figure 4J-M, cell cycle distribution using flow cytometry should be assessed in an additional cell line.
6. In line 271, the authors described that PROX1 is associated with an increase in DHCHR7. However, in the next paragraph they evaluated the effect of silencing HMGCR. Is this enzyme also increased? Please clarify.
7. The authors show that cholesterol biosynthesis is crucial in RMS. Would overexpression of the DHCR7 in shDHCR7 cells rescue the anti-tumor effects? A rescue experiment would give information on whether this enzyme has a direct role in driving RMS cell behavior.

****Minor comments:****

1. In line 287 "Supplementary Fig.1G and 1H" are mentioned, while it should be "Supplementary Fig.1I and 1J" since it regards the treatment with lovastan.
2. In line 340, authors mentioned the data "Supplementary Figure 4A and 4E", but there is not any corresponding data available in the Supplementary Information.
3. In the Legend of Figure 2L, authors mention "PRXO-1 silencing", this should be corrected to "shDHCR7". Also, please change "l" to capital "L".
4. In Figure 5G-H, please provide the data regarding loading control in the Western blot, as well as the molecular weights of the proteins presented.
5. Please, include the information of what black, red etc refer to in each Figure. This information is missing in several figures including Figure 2D, 2K, 3C, 3J, 3K, 3L which makes it difficult to follow.
6. The authors should indicate the numbers of biological replicates in individual experiments through whole figure legends.

2. Significance:

Significance (Required)

A nice and coherent study. Please see text above.

3. How much time do you estimate the authors will need to complete the suggested revisions:

Estimated time to Complete Revisions (Required)

(Decision Recommendation)

Between 1 and 3 months

Yes

Revision Plan

Manuscript number: RC-2024-02836R

Corresponding author(s): Riikka, Kivelä

1. General Statements

We thank the reviewers for their thoughtful and detailed feedback, which we found highly constructive and encouraging. The comments have been invaluable in guiding improvements to the clarity, rigor, and impact of our manuscript. Below, we provide our responses and outline the specific revisions we plan to make in response to each point raised. It was extremely encouraging that all the comments were highly relevant to the study demonstrating careful work by experts in the field and they truly help to improve the clarity and message of the manuscript.

2. Description of the planned revisions

Reviewer #1 (Evidence, reproducibility and clarity (Required)):

The manuscript by Gizaw et al characterizes the cholesterol biosynthetic pathway and the effect of its knockdown or inhibition on rhabdomyosarcoma tumor properties. The Authors find that the PROX1 transcription factor mediated cholesterol biosynthesis regulates rhabdomyosarcoma cell growth and proliferation. Blocking the cholesterol biosynthetic pathway leads to reduced proliferation, cell cycle arrest and ER-stress mediated enhanced apoptosis. Detailed transcriptomic analysis indicate gene expression patterns that support these findings.

Reviewer #1 (Significance (Required)):

Based on my expertise on rhabdomyosarcoma tumors, the manuscript is clear, concise and provides a significant advance to the field. Detailed mechanistic characterization is lacking, which takes away some of the significance of the findings, but the work done stands alone as description of the effect of the cholesterol biosynthetic pathway in rhabdomyosarcoma. Another aspect to be considered by the Authors is the potential specificity of targeting a ubiquitous pathway such as cholesterol biosynthesis, which is important to most cells and not only cancer cells. Overall, the manuscript may be revised to address the specific comments below.

Responses to Reviewer #1 comments

We thank the reviewer for the thoughtful and encouraging comments on our manuscript. We appreciate the recognition of the significance of our findings and the detailed suggestions provided. We are committed to addressing each of the reviewer's points to strengthen the manuscript and ensure clarity and rigor. Below, we outline how we plan to address each comment.

Major Comments:

1. Details of the healthy human myoblasts that are used in Figure 1A are not provided and should be updated. Evidence of PROX1 knockdown should be presented. What kind of pathways and gene ontology predictions were associated with the 225 genes that are commonly downregulated between all three cell lines in Figure 1A?

Response:

In the revised manuscript, we will include complete information regarding the origin and characterization of the healthy human myoblasts used in the Figure 1A. We will also provide additional data confirming PROX1 knockdown. Furthermore, we will present more details on the gene ontology (GO) and pathway enrichment analyses, and include the full results as supplemental data to highlight key biological processes affected by PROX1 silencing.

2. In Figure 2, while the effect of the shRNAs targeting DHCR7 or the DHCR7 inhibitor AY9944 are striking, it is not clear whether these effects are specific to rhabdomyosarcoma cells or cancer cells. A control, human myoblast cell line or another non-cancerous cell line should be used to repeat these experiments quantifying Caspase3/7 activity, cell growth etc. to assess the cancer cell specificity of such treatments. Evidence of DHCR7 knockdown at the protein level would add to the study.

Response:

We fully agree with the reviewer's suggestion and will conduct additional experiments using non-cancerous human myoblasts to assess the specificity of DHCR7 inhibition. These will include assays for Caspase 3/7 activation, cell viability, and proliferation under similar conditions. We have already performed western blot validation of DHCR7 knockdown at the protein level in RMS cell lines and will include this data in the manuscript. We will also highlight in the discussion that RMS cells in our experiments were highly vulnerable when cultured with full media (incl. FBS), whereas previous studies with breast cancer cells have shown that their growth is affected by cholesterol biosynthesis inhibition only if they are cultured without serum (containing cholesterol). We also show that cholesterol supplementation does not rescue RMS cells demonstrating the essential role of de novo cholesterol synthesis.

3. Western blots for Caspase3 quantification and a cell proliferation marker such as Cyclin D in shSCR and shDHCR7 tumor lysates would validate the data shown in the Figure 3. Are the shRNA constructs used inducible ones? If not, how do the Authors distinguish the effect of shDHCR7 on tumor engraftment versus tumor proliferation and growth? Many of the graphs need proper labeling of the axes and what the bars represent.

Response:

We will include western blot analysis for cleaved Caspase 3 and Cyclin D1 in tumor lysates to

support the observed effects on apoptosis and proliferation. We will clarify in the revised manuscript that the shRNA constructs used were constitutive. To distinguish between effects on tumor engraftment versus tumor growth, we will provide additional detail on how we controlled for initial cell viability and engraftment potential prior to injection. We will also revise figure panels to ensure all axes and error bars are clearly labeled.

4. Gene ontology and pathway analysis will add to Figure 4.

Response:

We will expand Figure 4 to include GO and pathway enrichment analyses of the RNA-seq data following DHCR7 knockdown. This will help illustrate the functional significance of the transcriptional changes and further support our conclusions regarding ER stress, apoptosis, and cell cycle regulation.

5. In Figure 5A, how do the Authors explain the upregulation of cholesterol biosynthetic pathway genes upon shDHCR7 treatment? Are these effects seen at the protein level and if alternate pathways maintain cholesterol biosynthesis, how do the Authors think this strategy will be viable to treat such tumors? In Figure 5G-H, was a loading control used? If so, blots for that should be included.

Response:

We will expand the discussion to address the compensatory transcriptional upregulation of cholesterol biosynthesis genes following DHCR7 knockdown, likely driven by SREBP-mediated feedback regulation. To support this, we will include western blot data for key enzymes in the pathway. We will also clarify that despite this transcriptional compensation, functional cholesterol synthesis is impaired due to DHCR7 silencing, which cannot be rescued by increased upstream pathway activity. Regarding Figure 5G–H, we will include the missing loading control images in the revised version. Protein normalization was performed using Stain-Free technology, which enables the quantification of total protein in each lane, and was analyzed using ImageLab 6.0.1 software (Bio-Rad). We will include the Stain-Free gel images to demonstrate equal protein loading and will also indicate the molecular weights of the presented proteins in the updated figure legend.

6. Lines 286–287 refer to Figure S1G, H; it should be corrected to Figure S1I, J.

Response:

We thank the reviewer for pointing this out. We will correct the figure citation in the revised manuscript.

Significance:

We will strengthen the Discussion by clarifying the rationale for targeting cholesterol biosynthesis in RMS, noting the apparent differential dependency in cancer versus normal cells. Future directions will include exploration of inducible systems and combination therapies for

improved tumor specificity.

Reviewer #2 (Evidence, reproducibility and clarity (Required)):

In this manuscript entitled "Targeting de novo cholesterol synthesis in rhabdomyosarcoma induces cell cycle arrest and triggers apoptosis through ER stress-mediated pathways" Gizaw et al investigate the crucial effect of targeting cholesterol biosynthesis in RMS. While this manuscript gives novel insights into putative therapeutic approach, there are some comments that should be address by the authors.

Reviewer #2 (Significance (Required)):

A nice and coherent study. Please see text above.

Response to Reviewer #2

We are grateful to the reviewer for the thoughtful and constructive comments on our manuscript. We appreciate your recognition of the novelty and therapeutic potential of our findings, and we thank you for highlighting specific areas that will help further improve the clarity, rigor, and reproducibility of our work. Below, we respond point-by-point to your comments and outline how we plan to address each issue in the revised version of the manuscript.

Major Comments:

1. The authors demonstrated a correlation between PROX1 levels and the cholesterol synthesis pathway. Which genes from the pathway are mostly affected? The manuscript could benefit from a graphical representation of the pathway showing up- and downregulated genes from the RNA-seq analysis. This will help in understanding why the authors decided to study HMGCR silencing as shown in Supplementary Figure 1A.

Response:

We fully agree and will include a new graphical figure showing the cholesterol biosynthesis pathway, with up- and downregulated genes from our RNA-seq data visually mapped. This is, indeed, interesting as the whole pathway is consistently downregulated. We chose to study specifically these two rate-limiting genes in the pathway, as DHCR7 is the last enzyme in the mevalonate pathway and its inhibition does not affect other arms deviating from this pathway. It was also recently found to be highly upregulated in pancreatic cancer, suggesting its role in cancer development/growth. HMGCR was chosen as it is the target for statins, which are widely used in treating high cholesterol and shown to be rather safe in clinical use. We will add this rationale to the manuscript to clarify our focus on HMGCR and DHCR7.

2. Based on the previous comment, are the genes from the cholesterol synthesis identified in the RNA-seq similar to those detected in the publicly available data set

presented in Figure 1E? In addition, validation of changes of these genes should be performed in the RMS cell lines as well as in myoblasts.

Response:

Yes, there is a significant overlap between the cholesterol biosynthesis genes identified in our RNA-seq dataset and those from the public dataset in Figure 1E. In the revised version, we will include this comparative analysis with the inclusion of the schematic figure (see our response #1). We also plan to perform qPCR validation of several key cholesterol biosynthesis genes in additional RMS cell lines and healthy myoblasts to reinforce the disease-specific regulation of this pathway.

3. In Figure 3, the authors study the impact of DHCR7-silencing in tumor growth in vivo. Please, provide stainings also for DHCR7 to show that cells indeed have silenced DHCR7.

Response:

Thank you for this important suggestion. We will include immunofluorescence staining for DHCR7 in xenograft tumor sections to confirm DHCR7 knockdown in vivo and visually validate the efficiency of our silencing strategy. We will also add qPCR results from the cells at the time when they were implanted confirming the deletion.

4. In Figure 4, the RNA-seq data revealed downregulation in E2F genes as well as genes involved in cell cycle progression. It would be important that the authors provide examples of these genes and validate this data by performing qPCR.

Response:

We will select representative cell cycle-related genes, including members of the E2F family and other G1/S and G2/M regulators, for qPCR validation in RMS cells following DHCR7 knockdown. Comparison to healthy myoblasts will be also performed. This will further substantiate the transcriptomic findings.

5. In Figure 4J–M, cell cycle distribution using flow cytometry should be assessed in an additional cell line.

Response:

We will repeat the flow cytometry-based cell cycle analysis in an additional RMS cell line to ensure reproducibility and confirm the generalizability of the observed G2/M arrest phenotype.

6. In line 271, the authors described that PROX1 is associated with an increase in DHCR7. However, in the next paragraph they evaluated the effect of silencing HMGCR. Is this enzyme also increased? Please clarify.

Response:

We appreciate the need for clarity. HMGCR expression is also elevated in RMS cells and regulated by PROX1. We will clarify this in the revised manuscript and update the text to explain the rationale behind examining both enzymes: HMGCR as the rate-limiting enzyme at the top of the cholesterol biosynthesis pathway, and DHCR7 as the final step enzyme. See also our response to question #1.

7. The authors show that cholesterol biosynthesis is crucial in RMS. Would overexpression of DHCR7 in shDHCR7 cells rescue the anti-tumor effects? A rescue experiment would give information on whether this enzyme has a direct role in driving RMS cell behavior.

Response:

This is an excellent suggestion. We are currently generating a DHCR7 rescue construct and plan to perform these experiments. While these data may not be available in time for the current revision, we will clearly outline this approach as a key next step in our *Discussion* section and incorporate results if available.

Minor Comments:

1. In line 287 "Supplementary Fig.1G and 1H" are mentioned, while it should be "Supplementary Fig.1I and 1J" since it regards the treatment with lovastatin.

Response:

Thank you for catching this. We will correct the figure references accordingly.

2. In line 340, authors mentioned the data "Supplementary Figure 4A and 4E", but there is not any corresponding data available in the Supplementary Information.

Response:

We apologize for this oversight. These references will be corrected, and any missing supplementary data will be properly included and labeled.

3. In the Legend of Figure 2L, authors mention "PRXO-1 silencing", this should be corrected to "shDHCR7". Also, please change "I" to capital "L".

Response:

This will be corrected in the revised figure legend.

4. In Figure 5G–H, please provide the data regarding loading control in the Western blot, as well as the molecular weights of the proteins presented.

Response:

We thank the reviewer for this important point. For the Western blot analysis in Figure 5G–H,

normalization was performed by quantifying the total protein in each lane using Bio-Rad's Stain-Free technology and analyzed with ImageLab 6.0.1 software. This approach allows for accurate lane-to-lane comparison without relying on a single housekeeping protein. We will add the Stain-Free total protein images as a supplemental figure (Supplementary Figure) and include the molecular weights for each of the proteins in the figure legend to improve clarity and reproducibility.

5. Please, include the information of what black, red etc refer to in each figure. This information is missing in several figures including Figure 2D, 2K, 3C, 3J, 3K, 3L which makes it difficult to follow.

Response:

We agree and will update all relevant figure legends to clearly explain color coding, symbols, and what each bar or line represents to improve figure clarity.

6. The authors should indicate the numbers of biological replicates in individual experiments throughout whole figure legends.

Response:

Thank you for the suggestion. We will include the number of biological replicates for each experiment in the figure legends to enhance transparency and reproducibility.

Significance:

We sincerely appreciate the reviewer's positive evaluation of the significance of our study. Your encouraging feedback reinforces the value of our work, and we are motivated to further strengthen the manuscript based on your detailed and constructive suggestions above. We are confident that with the planned revisions and additional validations, our study will provide an even clearer and more impactful contribution to the understanding and therapeutic targeting of cholesterol biosynthesis in rhabdomyosarcoma.

Additional new data:

While the manuscript was under review, we continued experiments and analyses using human RMS patient samples available in public data repositories. We re-analyzed an RMS sc/snCell RNA-seq atlas consisting of 107 523 cells/nuclei from primary RMS tumors, metastases, PDX and cells. The human patient results strongly support our findings from cell culture and in vivo animal studies, as the results demonstrate that those RMS cells that have high DHCR7 and HMGCR expression have increased mTORC signaling as well as cell cycle and MYC activity (see table below). This was true for the primary tumors, metastasis, cell lines and primary cultures and for PDX. Thus, targeting of these cells by inhibiting cholesterol biosynthesis could be highly beneficial for cancer treatment. We will include a new figure from these results in the manuscript for added translational relevance.

Revision Plan

+: upregulated, FDR < 0.05 (+): upregulated, FDR > 0.05	MTORC1 signaling	Androgen response	Estrogen early response	Estrogen late response	Myc V1 targets	Myc V2 targets	G2M	E2F
Cell lines and primary cultures	+	+	+	(+)			+	+
Patient model, primary tumors	+	+	+	+	+	+	+	+
Patient model, metastasis	+	+	(+)	(-)			(+)	(+)
PDX	+	(+)		(+)	+	+	+	+

In addition, we performed Kaplan-Meier survival analyses using sarcoma patient data from 259 patients including rhabdomyosarcoma. This data shows significantly reduced survival in patients with high DHCR7 and HMGR expression (see images below). Of note, for many other cancer types, e.g. breast cancer, there was no significant correlation, and if something, it goes into opposite direction, indicating specificity to sarcomas. This data will be added as supplemental figure to the manuscript.

Revision Plan

3. Description of the revisions that have already been incorporated in the transferred manuscript
4. Description of analyses that authors prefer not to carry out

29th Apr 2025

Dear Prof. Kivelä,

Thank you for the submission of your research manuscript to our editorial offices. I have now had the opportunity to read the manuscript, the referee reports and your point-by-point response and to discuss it with the other members of our editorial team. We all agreed that the manuscript fits the scope of EMBO Molecular Medicine but also that the referees raised serious concerns that should be addressed in major revision. In our opinion your revision plan addresses the referees' criticism adequately and therefore we would like to invite submission of the revised manuscript to our journal.

We would welcome the submission of a revised version within three months for further consideration. Please let us know if you require longer to complete the revision.

Please use this link to login to the manuscript system and submit your revision: <https://embomolmed.msubmit.net/cgi-bin/main.plex>

I look forward to receiving your revised manuscript.

Yours sincerely,

Zeljko Durdevic

Zeljko Durdevic
Senior Editor
EMBO Molecular Medicine

We require:

- 1) A .docx formatted version of the manuscript text (including legends for main figures, EV figures and tables). Please make sure that the changes are highlighted to be clearly visible.
- 2) Individual production quality figure files as .eps, .tif, .jpg (one file per figure). For guidance, download the 'Figure Guide PDF': (<https://www.embopress.org/page/journal/17574684/authorguide#figureformat>).
- 3) A .docx formatted letter INCLUDING the reviewers' reports and your detailed point-by-point responses to their comments. As part of the EMBO Press transparent editorial process, the point-by-point response is part of the Review Process File (RPF), which will be published alongside your paper.
- 4) A complete author checklist, which you can download from our author guidelines (<https://www.embopress.org/page/journal/17574684/authorguide#submissionofrevisions>). Please insert information in the checklist that is also reflected in the manuscript. The completed author checklist will also be part of the RPF.
- 5) Please note that all corresponding authors are required to supply an ORCID ID for their name upon submission of a revised manuscript.
- 6) It is mandatory to include a 'Data Availability' section after the Materials and Methods. Before submitting your revision, primary

datasets produced in this study need to be deposited in an appropriate public database, and the accession numbers and database listed under 'Data Availability'. Please remember to provide a reviewer password if the datasets are not yet public (see <https://www.embopress.org/page/journal/17574684/authorguide#dataavailability>).

12) Author contributions: the contribution of every author must be detailed in a separate section (before the acknowledgments).

13) A Conflict of Interest statement should be provided in the main text.

14) Every published paper now includes a 'Synopsis' to further enhance discoverability. Synopses are displayed on the journal webpage and are freely accessible to all readers. They include a short stand first (maximum of 300 characters, including space) as well as 2-5 one-sentences bullet points that summarizes the paper. Please write the bullet points to summarize the key NEW findings. They should be designed to be complementary to the abstract - i.e. not repeat the same text. We encourage inclusion of key acronyms and quantitative information (maximum of 30 words / bullet point). Please use the passive voice. Please attach these in a separate file or send them by email, we will incorporate them accordingly.

15) Include a Reagents and Tools Table as part of the Methods section, which can be downloaded from our author guidelines (<https://www.embopress.org/page/journal/17574684/authorguide#structuredmethods>)

Rev_Com_number: RC-2024-02836

New_manu_number: EMM-2025-21866-T

Corr_author: Kivelä

Title: Targeting cholesterol synthesis in rhabdomyosarcoma induces cell cycle arrest and triggers apoptosis

Revision Plan

Manuscript number: RC-2024-02836R
Corresponding author(s): Riikka Kivelä

1. General Statements

We thank the reviewers for their thoughtful and detailed feedback, which we found highly constructive and encouraging. The comments have been valuable to improve the clarity, rigor, and impact of our manuscript. Below, we provide our responses and outline the specific revisions we have made. It was very encouraging that the comments were highly relevant to the study demonstrating careful work by experts in the field and they truly helped to improve the message of the manuscript.

2. Description of the planned revisions

Reviewer #1 (Evidence, reproducibility and clarity (Required)):

The manuscript by Gizaw et al characterizes the cholesterol biosynthetic pathway and the effect of its knockdown or inhibition on rhabdomyosarcoma tumor properties. The Authors find that the PROX1 transcription factor mediated cholesterol biosynthesis regulates rhabdomyosarcoma cell growth and proliferation. Blocking the cholesterol biosynthetic pathway leads to reduced proliferation, cell cycle arrest and ER-stress mediated enhanced apoptosis. Detailed transcriptomic analysis indicate gene expression patterns that support these findings.
Reviewer #1 (Significance (Required)):

Based on my expertise on rhabdomyosarcoma tumors, the manuscript is clear, concise and provides a significant advance to the field. Detailed mechanistic characterization is lacking, which takes away some of the significance of the findings, but the work done stands alone as description of the effect of the cholesterol biosynthetic pathway in rhabdomyosarcoma. Another aspect to be considered by the Authors is the potential specificity of targeting a ubiquitous pathway such as cholesterol biosynthesis, which is important to most cells and not only cancer cells. Overall, the manuscript may be revised to address the specific comments below.

Responses to Reviewer #1 comments

We thank the reviewer for the thoughtful and encouraging comments on our manuscript. We appreciate the recognition of the significance of our findings and the detailed suggestions provided.

Major Comments:

Revision Plan

1. Details of the healthy human myoblasts that are used in Figure 1A are not provided and should be updated. Evidence of PROX1 knockdown should be presented. What kind of pathways and gene ontology predictions were associated with the 225 genes that are commonly downregulated between all three cell lines in Figure 1A?

Response:

In the revised manuscript, we have included complete information regarding the origin and characterization of the healthy human myoblasts used in the Figure 1A and in the revision experiments in the Figure 2 and 2EV. We have also added the PROX1 silencing levels to the Figure 1A. Furthermore, gene ontology (GO) analysis of the 225 genes commonly downregulated across all three RMS cell lines revealed no significantly enriched biological processes (BP), but did identify enrichment of the cellular component (CC) terms 'mitochondrion' and 'mitochondrial matrix. This is now added to the results.

Commented [MK1]: Need to be added

Commented [NG2R1]: Yes please use final figure

2. In Figure 2, while the effect of the shRNAs targeting DHCR7 or the DHCR7 inhibitor AY9944 are striking, it is not clear whether these effects are specific to rhabdomyosarcoma cells or cancer cells. A control, human myoblast cell line or another non-cancerous cell line should be used to repeat these experiments quantifying Caspase3/7 activity, cell growth etc. to assess the cancer cell specificity of such treatments. Evidence of DHCR7 knockdown at the protein level would add to the study.

Response:

Thank you for excellent suggestion. We have now performed these live cell imaging experiments with healthy primary human myoblasts and human to assess the specificity of DHCR7 inhibition. We also confirmed DHCR7 knockdown at the protein level in RMS cell lines and this data is now included in the Figure 4J. We also highlight in the discussion that RMS cells in our experiments were highly vulnerable even when cultured in full media containing FBS (and thus cholesterol), whereas previous studies in other cancer types have shown that growth is affected by cholesterol biosynthesis inhibition only under serum-free conditions, since those cells can compensate via LDL-cholesterol uptake (as discussed on page 10). Importantly, we also show that LDL-cholesterol supplementation did not rescue RMS cells demonstrating the essential role of de novo cholesterol synthesis (Figure 2A-B).

3. Western blots for Caspase3 quantification and a cell proliferation marker such as Cyclin D in shSCR and shDHCR7 tumor lysates would validate the data shown in the Figure 3. Are the shRNA constructs used inducible ones? If not, how do the Authors distinguish the effect of shDHCR7 on tumor engraftment versus tumor proliferation and growth? Many of the graphs need proper labeling of the axes and what the bars represent.

Response:

Unfortunately we did not have protein samples from the tumors, but the expression levels of these proteins have been estimated based on the IHC images (Figure 3). In addition, we have

Revision Plan

now performed qPCR from the tumors to confirm the findings that were obtained from cell cultures and IHC (Figure 3). To further validate that DHCR7 silencing leads to a reduction in Cyclin D1 expression, we performed western blot analysis on cell lysates, as shown in Figure 4J. We have also clarified in the manuscript text that the shRNA constructs used were constitutive (Methods, page 14). The cells were injected to the mice immediately after silencing to ensure their viability at the time of injection and engraftment (added to the Methods, page 15). Thank you for pointing the defects in figure labeling. We have now revised the figure panels and legends to ensure all axes and error bars are clearly labeled.

4. Gene ontology and pathway analysis will add to Figure 4.

Response:

We have added the GO enrichment analyses results of the RNA-seq data following DHCR7 silencing to Figure EV4.

5. In Figure 5A, how do the Authors explain the upregulation of cholesterol biosynthetic pathway genes upon shDHCR7 treatment? Are these effects seen at the protein level and if alternate pathways maintain cholesterol biosynthesis, how do the Authors think this strategy will be viable to treat such tumors? In Figure 5G-H, was a loading control used? If so, blots for that should be included.

Response:

We have added discussion on the compensatory transcriptional upregulation of cholesterol biosynthesis genes observed following DHCR7 knockdown, likely driven by SREBP-mediated feedback regulation. Similar feedback has been reported in other cancer cell types, but typically under conditions lacking extracellular LDL-derived cholesterol. To support this observation, we have performed western blot analyses to explore if this also occurs at protein level in RD and KLHEL1 cells (Figure EV4). We have also explained that despite this compensation, functional cholesterol synthesis is impaired due to DHCR7 silencing, and this cannot be rescued by increased upstream pathway activity. Regarding Figure 5G–H, we have now included the gels with the loading control in the source data. Protein normalization was performed using Stain-Free technology, which enables the quantification of total protein in each lane, and was analyzed using ImageLab 6.0.1 software (Bio-Rad). We have now included all the Stain-Free gel images to demonstrate equal protein loading and have added the molecular weights of the presented proteins in the updated figure legend. All original gels and their quantifications are provided in the source data.

6. Lines 286–287 refer to Figure S1G, H; it should be corrected to Figure S1I, J.

Response:

We thank the reviewer for pointing this out. We have updated the figure citation in the revised manuscript.

Revision Plan

Significance:

We have strengthened the Discussion by clarifying the rationale for targeting cholesterol biosynthesis in RMS, noting the apparent differential dependency in cancer versus normal cells. Future directions will include exploration of inducible systems and combination therapies for improved tumor specificity.

Reviewer #2 (Evidence, reproducibility and clarity (Required)):

In this manuscript entitled "Targeting de novo cholesterol synthesis in rhabdomyosarcoma induces cell cycle arrest and triggers apoptosis through ER stress-mediated pathways" Gizaw et al investigate the crucial effect of targeting cholesterol biosynthesis in RMS. While this manuscript gives novel insights into putative therapeutic approach, there are some comments that should be address by the authors.

Reviewer #2 (Significance (Required)):

A nice and coherent study. Please see text above.

Response to Reviewer #2

Thank you for the thoughtful and constructive comments on our manuscript. We appreciate your recognition of the novelty and therapeutic potential of our findings, and we thank you for highlighting specific areas that will help further improve the clarity, rigor, and reproducibility of our work.

Major Comments:

1. The authors demonstrated a correlation between PROX1 levels and the cholesterol synthesis pathway. Which genes from the pathway are mostly affected? The manuscript could benefit from a graphical representation of the pathway showing up- and downregulated genes from the RNA-seq analysis. This will help in understanding why the authors decided to study HMGCR silencing as shown in Supplementary Figure 1A.

Response:

Thank you for a great suggestion. We have now included a graphical representation, showing the cholesterol biosynthesis pathway and highlighted the rate-limiting enzymes that were targeted in our experiments (Figure 2A). Heatmaps in Figure 1 show the expression levels of the MVA-pathway enzymes in the PROX1 silenced cells compared to controls (Figure 1B) and the high expression of the same genes in different RMS cell lines vs healthy myoblasts (Figure 1F). This is, indeed, interesting as the whole pathway is consistently downregulated upon PROX1 silencing.

We chose to study specifically these two rate-limiting genes in the pathway. HMGCR was first studied as it is the target for statins, which are widely used in the treatment of patients with high

Revision Plan

cholesterol and shown to be rather safe in clinical use. DHCR7 is the last enzyme in the mevalonate pathway, and its inhibition does not affect the other arms deviating from this pathway. We have added this rationale to the manuscript to clarify our focus on HMGCR and DHCR7 (page 5).

2. Based on the previous comment, are the genes from the cholesterol synthesis identified in the RNA-seq similar to those detected in the publicly available data set presented in Figure 1E? In addition, validation of changes of these genes should be performed in the RMS cell lines as well as in myoblasts.

Response:

Yes, there is a significant overlap between the cholesterol biosynthesis genes identified in our RNA-seq dataset and those from the public dataset in Figure 1E. We have now performed qPCR validation of several key cholesterol biosynthesis genes in three different RMS cell lines (RD, RH30, KLHEL1) and compared the expression to healthy myoblasts to reinforce the disease-specific regulation of this pathway (Figure 1F).

3. In Figure 3, the authors study the impact of DHCR7-silencing in tumor growth in vivo. Please, provide stainings also for DHCR7 to show that cells indeed have silenced DHCR7.

Response:

Thank you for this important suggestion. We have now included immunofluorescence staining for DHCR7 in tumor sections to evaluate DHCR7 knockdown in vivo (Figure 3G,K and EV3). We have also added qPCR results from the cells at the time when they were implanted confirming the deletion (Figure EV3). Similarly to what we have seen previously in PROX1 silenced tumors (Gizaw et al, PNAS 2022), the data indicate that escape clones with lower silencing efficiency have formed a majority of the tumor as the silencing efficiency is lower in tumors than in the implanted cells.

4. In Figure 4, the RNA-seq data revealed downregulation in E2F genes as well as genes involved in cell cycle progression. It would be important that the authors provide examples of these genes and validate this data by performing qPCR.

Response:

We have now performed qPCR and western blotting for selected cell cycle genes/proteins in three RMS cell lines. The results have are included in Figure 4J and 4K. From the GSEA results, one can see that almost all E2F target genes have been downregulated (each black bar represents one gene and they are mostly on the blue (=downregulated) side of the plot).

5. In Figure 4J–M, cell cycle distribution using flow cytometry should be assessed in an additional cell line.

Revision Plan

Response:

We have now repeated the flow cytometry-based cell cycle analysis in RH30 cells to ensure reproducibility and confirm the generalizability of the observed G2/M arrest phenotype (Figure 4L). The effect was even stronger in this FP-RMS cell line compared RD cells (FN-RMS).

6. In line 271, the authors described that PROX1 is associated with an increase in DHCR7. However, in the next paragraph they evaluated the effect of silencing HMGCR. Is this enzyme also increased? Please clarify.

Response:

We appreciate the need for clarity. HMGCR expression is also elevated in RMS cells and downregulated by PROX1 silencing. We have clarified this in the revised manuscript to explain the rationale behind examining both enzymes (page 5): HMGCR as the rate-limiting enzyme on the top of the cholesterol biosynthesis pathway, and DHCR7 is the final bottom enzyme. See also our response to question #1.

7. The authors show that cholesterol biosynthesis is crucial in RMS. Would overexpression of DHCR7 in shDHCR7 cells rescue the anti-tumor effects? A rescue experiment would give information on whether this enzyme has a direct role in driving RMS cell behavior.

Response:

This is an excellent suggestion. We are currently generating a DHCR7 overexpression construct and plan to perform these experiments. Unfortunately, this data is not available within the timeframe for the current manuscript revision. We have discussed the role of DHCR7 in the manuscript in the context of external cholesterol supplementation, which could not rescue the phenotype. Furthermore, DHCR7 silencing induced the compensatory expression of the upstream genes/proteins in the pathway, which cannot not improve the situation either when DHCR7 levels are low. We believe that fully functional MVA pathway is thus needed for RMS cells and by adding DHCR7 back would rescue the phenotype.

Commented [MK3]: Should we add a short comment on this?

Minor Comments:

1. In line 287 "Supplementary Fig.1G and 1H" are mentioned, while it should be "Supplementary Fig.1I and 1J" since it regards the treatment with lovastatin.

Response:

Thank you for noting this. We have corrected the figure references accordingly.

2. In line 340, authors mentioned the data "Supplementary Figure 4A and 4E", but there is not any corresponding data available in the Supplementary Information.

Revision Plan

Response:

We apologize for this oversight. We have now corrected the figure references and added new supplementary data.

3. In the Legend of Figure 2L, authors mention "PRXO-1 silencing", this should be corrected to "shDHCR7". Also, please change "I" to capital "L".

Response:

This has been corrected in the revised figure legend.

4. In Figure 5G–H, please provide the data regarding loading control in the Western blot, as well as the molecular weights of the proteins presented.

Response:

We thank the reviewer for this important point. For the Western blot analysis in Figure 5G–H, normalization was performed by quantifying the total protein in each lane using Bio-Rad's Stain-Free technology and analyzed with ImageLab 6.0.1 software. This approach allows for accurate lane-to-lane comparison without relying on a single housekeeping protein. We have added the Stain-Free total protein images and full gel images as source data and included the molecular weights for each of the proteins to improve clarity and reproducibility.

5. Please, include the information of what black, red etc refer to in each figure. This information is missing in several figures including Figure 2D, 2K, 3C, 3J, 3K, 3L which makes it difficult to follow.

Response:

We agree and have now updated all relevant figure legends to clearly explain color coding, symbols, and what each bar or line represents to improve figure clarity.

6. The authors should indicate the numbers of biological replicates in individual experiments throughout whole figure legends.

Response:

Thank you for the suggestion. We have included the number of biological replicates for each experiment in the figure legends.

Additional new data:

While the manuscript was under review, we continued experiments and analyses using human RMS patient samples available in public data repositories. We re-analyzed an RMS sc/snCell RNA-seq atlas consisting of 107 523 cells/nuclei from primary RMS tumors, metastases, PDX

Revision Plan

and cells. The human patient results strongly support our findings from cell culture and in vivo animal studies, as the results demonstrate that those RMS cells that have high DHCR7 and HMGCR expression have increased mTORC signaling as well as cell cycle and MYC activity. This was true for the primary tumors, metastasis, cell lines and primary cultures and for PDX. Thus, targeting of these cells by inhibiting cholesterol biosynthesis could be highly beneficial for cancer treatment. We have now included a new figure from these results in the manuscript for added translational relevance (Figure 6).

In addition, we performed Kaplan-Meier survival analyses using RMS patient data from 101 patients. This data showed reduced survival in patients with high DHCR7 and HMGCR expression. Of note, for many other cancer types, e.g. breast cancer, there was no significant correlation, and if something, it goes into opposite direction, indicating specificity to sarcomas. This data has been added to Figure 6.

30th Sep 2025

Dear Prof. Kivelä,

Thank you for the submission of your revised manuscript to EMBO Molecular Medicine. We have now heard back from the one referee who agreed to re-evaluate your manuscript. This referee also assessed author responses to concerns raised by other referees. I am pleased to inform you that we will be able to accept your manuscript pending the following final amendments:

- 1) Please address referee's minor points. Please note that it is not required to add line numbers to the point-by-point response at this stage.
- 2) Figures: Please remove legends from all figure files.
- 3) In the main manuscript file, please do the following:
 - Please address all comments suggested by our data editors listed below:
 - o Data availability statement:
 1. Please note that the specific URL for GSE279213 dataset is not provided in the data availability statement.
 - o Figure legends:
 1. Please define the annotated p values ****/****/* as well as provide the exact p-values for the same in the legend of figure 2B, C, F, G, H, K, L, M, N, O as appropriate.
 2. Please note that the exact p values are not provided in the legends of figures 1D, E; 2D, J; 3 A, C, L, M, N, O; 4K, L; 5F-H; EV1 A, B, D, E, F, H; EV3 A, B, K, L, M, N.
 3. Please indicate the statistical test used for data analysis in the legends of figures 1B, 2B, C, F, G, H, L, M, N, O; 6I; EV1 A, B, D, E, F, H; EV3 A, B, K, L, M, N; EV4 A.
 4. Please note that information related to n is missing in the legends of figures 1E; 2B, C, F, H, K, L-S; 3C, 5H, 6E-H; EV1 A, B, D, E, F, H, I, J; EV3 A, B, K, L, M, N.
 5. Please note that the error bars are not defined in the legends of figures 2B, C, E, F, G, H, J, K; L-S.
 - Add callouts for Fig 2B and Fig 4L.
 - In Methods, provide the statement that informed consent was obtained from all human subjects and that the experiments conformed to the principles set out in the WMA Declaration of Helsinki and the Department of Health and Human Services Belmont Report.
 - In Methods, a statistical paragraph should reflect all information that you have filled in the Authors Checklist, especially regarding randomization, blinding, replication.
 - In Methods, add the following paragraph:

Graphics:

(some of the... OR Figure #... OR synopsis) Graphics were created with BioRender.com.

- Indicate in legends exact n and exact p values, not a range, along with the statistical test used. To keep the figures "clear" some authors found providing an Appendix table Sx with all exact p-values preferable. You are welcome to do this if you want to.
- Author contributions: Please remove it from the manuscript and specify author contributions in our submission system. CRediT has replaced the traditional author contributions section because it offers a systematic machine-readable author contributions format that allows for more effective research assessment. You are encouraged to use the free text boxes beneath each contributing author's name to add specific details on the author's contribution. More information is available in our guide to authors: <https://www.embopress.org/page/journal/17574684/authorguide#authorshipguidelines>
- In data availability statement please remove the first and the last sentence and only leave information about RNA sequencing data. Please use the following format to report the accession number of your data:

[data type]: [full name of the resource] [accession number/identifier] ([doi or URL or identifiers.org/DATABASE:ACCESSION])

Please check "Author Guidelines" for more information.

<https://www.embopress.org/page/journal/17574684/authorguide#availabilityofpublishedmaterial>

- 4) Tables: Rename Tables EV1 and EV2 to "Dataset EV1" and "Dataset EV2" and the remaining table to "Table EV1". Please correct table nomenclature to Table EV1 in the manuscript.
- 5) Reagent table: Please correct the citation of "Appendix Table" to "Table EV1".
- 6) Funding: If available, please add project numbers in the manuscript and our submission system.
- 7) Source data: Please upload the source data as one zipped file per figure and provide completed source data checklist.
- 8) Synopsis:
 - Synopsis image: Please upload the image as a high-resolution jpeg file 550 px-wide x 300-600 pixels high.
 - Please check your synopsis text and image before submission with your revised manuscript. Please be aware that in the proof stage minor corrections only are allowed (e.g., typos).
- 9) As part of the EMBO Publications transparent editorial process initiative (see our Editorial at <http://embomolmed.embopress.org/content/2/9/329>), EMBO Molecular Medicine will publish online a Review Process File (RPF)

to accompany accepted manuscripts. This file will be published in conjunction with your paper and will include the anonymous referee reports, your point-by-point response and all pertinent correspondence relating to the manuscript. Let us know whether you agree with the publication of the RPF and as here, if you want to remove or not any figures from it prior to publication. Please note that the Authors checklist will be published at the end of the RPF.

10) Please provide a point-by-point letter INCLUDING my comments as well as the reviewer's reports and your detailed responses (as Word file).

I look forward to reading a new revised version of your manuscript as soon as possible.

Yours sincerely,

Zeljko Durdevic

Zeljko Durdevic
Senior Editor
EMBO Molecular Medicine

*** Instructions to submit your revised manuscript ***

To submit your manuscript, please follow this link:

<https://embomolmed.msubmit.net/cgi-bin/main.plex>

- 1) a .docx formatted version of the manuscript text (including Figure legends and tables)
- 2) Separate figure files*
- 3) supplemental information as Expanded View and/or Appendix. Please carefully check the authors guidelines for formatting Expanded view and Appendix figures and tables at <https://www.embopress.org/page/journal/17574684/authorguide#expandedview>
- 4) a letter INCLUDING the reviewer's reports and your detailed responses to their comments (as Word file).
- 5) The paper explained: EMBO Molecular Medicine articles are accompanied by a summary of the articles to emphasize the major findings in the paper and their medical implications for the non-specialist reader. Please provide a draft summary of your article highlighting
 - the medical issue you are addressing,
 - the results obtained and
 - their clinical impact.This may be edited to ensure that readers understand the significance and context of the research. Please refer to any of our published articles for an example.

6) Author contributions: the contribution of every author must be detailed in a separate section.

7) EMBO Molecular Medicine now requires a complete author checklist (<https://www.embopress.org/page/journal/17574684/authorguide>) to be submitted with all revised manuscripts. Please use the checklist as guideline for the sort of information we need WITHIN the manuscript. The checklist should only be filled with page

numbers were the information can be found. This is particularly important for animal reporting, antibody dilutions (missing) and exact values and n that should be indicated instead of a range.

8) Every published paper now includes a 'Synopsis' to further enhance discoverability. Synopses are displayed on the journal webpage and are freely accessible to all readers. They include a short stand first (maximum of 300 characters, including space) as well as 2-5 one sentence bullet points that summarise the paper. Please write the bullet points to summarise the key NEW findings. They should be designed to be complementary to the abstract - i.e. not repeat the same text. We encourage inclusion of key acronyms and quantitative information (maximum of 30 words / bullet point). Please use the passive voice. Please attach these in a separate file or send them by email, we will incorporate them accordingly.

You are also welcome to suggest a striking image or visual abstract to illustrate your article. If you do please provide a jpeg file 550 px-wide x 300-600px high.

9) A Conflict of Interest statement should be provided in the main text

10) Please note that we now mandate that all corresponding authors list an ORCID digital identifier. This takes <90 seconds to complete. We encourage all authors to supply an ORCID identifier, which will be linked to their name for unambiguous name identification.

Currently, our records indicate that the ORCID for your account is 0000-0002-2686-8890.

Link Not Available

11) Include a Reagents and Tools Table as part of the Methods section, which can be downloaded from our author guidelines (<https://www.embopress.org/page/journal/17574684/authorguide#structuredmethods>)

Photos 400-800 DPI

*Additional important information regarding figures and illustrations can be found at

<https://bit.ly/EMBOPressFigurePreparationGuideline>. See also figure legend preparation guidelines:

<https://www.embopress.org/page/journal/17574684/authorguide#figureformat>

***** Reviewer's comments *****

Referee #1 (Comments on Novelty/Model System for Author):

The manuscript has improved significantly following revision.

Referee #1 (Remarks for Author):

The Authors have revised the manuscript and addressed most of the comments. Also, new data has been added to the manuscript, unrelated to the Reviewer comments, which adds to the manuscript significantly. Overall, the revision appears reasonably done, but a few issues are still unclear, which should be addressed.

1. Please mention clearly the page and line numbers for the text and page/figure number along with the panel number for the figures, regarding the revision work that has been carried out.

While some of the areas in the text are in a different colored font to mark the changes made, it is unclear which part refers to what comment, and not all changes are marked. For example, the changes made in the materials and methods for the human myoblast information has not been marked in the revised manuscript (Comment 1 by Reviewer 1). Also, the -log values added to Figure 1A should be explained in the legend and the text.

Similarly, in response to Comment 1 by Reviewer 2, the Authors respond "DHCR7 is the last enzyme in the mevalonate pathway, and its inhibition does not affect the other arms deviating from this pathway. We have added this rationale to the manuscript to clarify our focus on HMGCR and DHCR7 (page 5)". It is not clear which sentences on page 5 of the revised

manuscript the Authors are referring to as the rationale.

In response to Comment 3 by Reviewer 1, the Authors say "Unfortunately we did not have protein samples from the tumors, but the expression levels of these proteins have been estimated based on the IHC images (Figure 3). In addition, we have now performed qPCR from the tumors to confirm the findings that were obtained from cell cultures and IHC (Figure 3)"; please mention which panels the Authors are referring to in Figure 3.

Unless the line numbers and figure numbers with panels are provided clearly for each comment in the response to reviewer comments, it is not possible to evaluate the changes made during revision.

2. The Authors have not addressed part of Comment 3 by Reviewer 1 regarding distinguishing between the effect of shDHRC7 on tumor engraftment versus tumor proliferation and growth. I understand that generating an inducible shRNA construct to test this could take a long time and therefore, suggest that the Authors add a sentence saying something to the effect of "not having tested to distinguish between tumor engraftment versus tumor proliferation and growth in the xenograft experiments".

3. For Comment 4 by Reviewer 2, the Authors mention "From the GSEA results, one can see that almost all E2F target genes have been downregulated (each black bar represents one gene and they are mostly on the blue (=downregulated) side of the plot". Which GSEA results are the Authors referring to? Clarify the figure and panel, along with the text line numbers.

4. For Comment 6 by Reviewer 2, the Authors mention "HMGR expression is also elevated in RMS cells and downregulated by PROX1 silencing. We have clarified this in the revised manuscript to explain the rationale behind examining both enzymes (page 5)". Specify the line numbers in the revised manuscript which addresses this.

5. For Comment 7 by Reviewer 2, the Authors respond "This is an excellent suggestion. We are currently generating a DHCR7 overexpression construct and plan to perform these experiments. Unfortunately, this data is not available within the timeframe for the current manuscript revision. We have discussed the role of DHCR7 in the manuscript in the context of external cholesterol supplementation, which could not rescue the phenotype. Furthermore, DHCR7 silencing induced the compensatory expression of the upstream genes/proteins in the pathway, which cannot not improve the situation either when DHCR7 levels are low. We believe that fully functional MVA pathway is thus needed for RMS cells and by adding DHCR7 back would rescue the phenotype". Are the Authors saying there is improvement with "Cannot not improve"? If the DHCR7 construct is not ready, can the Authors discuss this query raised by the Reviewer and its significance in the Discussion?

Rev_Com_number: RC-2024-02836

New_manu_number: EMM-2025-21866-V2

Corr_author: Kivelä

Title: Inhibiting cholesterol synthesis halts rhabdomyosarcoma growth via ER stress and cell cycle arrest

Responses to reviewer's and editors comments

Thank you for the submission of your revised manuscript to EMBO Molecular Medicine. We have now heard back from the one referee who agreed to re-evaluate your manuscript. This referee also assessed author responses to concerns raised by other referees. I am pleased to inform you that we will be able to accept your manuscript pending the following final amendments:

I look forward to reading a new revised version of your manuscript as soon as possible.

Yours sincerely,

Zeljko Durdevic

Zeljko Durdevic
Senior Editor
EMBO Molecular Medicine

Reviewer's comments and our responses

***** Reviewer's comments *****

Referee #1 (Comments on Novelty/Model System for Author):

The manuscript has improved significantly following revision.

Referee #1 (Remarks for Author):

The Authors have revised the manuscript and addressed most of the comments. Also, new data has been added to the manuscript, unrelated to the Reviewer comments, which adds to the manuscript significantly. Overall, the revision appears reasonably done, but a few issues are still unclear, which should be addressed.

1. Please mention clearly the page and line numbers for the text and page/figure number along with the panel number for the figures, regarding the revision work that has been carried out.

While some of the areas in the text are in a different colored font to mark the changes made, it is unclear which part refers to what comment, and not all changes are marked. For example, the changes made in the materials and methods for the human myoblast information has not been marked in the revised manuscript (Comment 1 by Reviewer 1). Also, the $-\log$ values added to Figure 1A should be explained in the legend and the text.

We are sorry for the confusion regarding the changes that were made in our revision. We have now added more information for the human myoblasts as asked

by the data editor. Explanation on the $-\log_2$ FC values in Figure 1A have been added to the legend and the results.

Similarly, in response to Comment 1 by Reviewer 2, the Authors respond "DHCR7 is the last enzyme in the mevalonate pathway, and its inhibition does not affect the other arms deviating from this pathway. We have added this rationale to the manuscript to clarify our focus on HMGCR and DHCR7 (page 5)". It is not clear which sentences on page 5 of the revised manuscript the Authors are referring to as the rationale.

This can be found in lines 159-164.

In response to Comment 3 by Reviewer 1, the Authors say "Unfortunately we did not have protein samples from the tumors, but the expression levels of these proteins have been estimated based on the IHC images (Figure 3). In addition, we have now performed qPCR from the tumors to confirm the findings that were obtained from cell cultures and IHC (Figure 3)"; please mention which panels the Authors are referring to in Figure 3.

Unless the line numbers and figure numbers with panels are provided clearly for each comment in the response to reviewer comments, it is not possible to evaluate the changes made during revision.

We are sorry for this problem, these data can be found in the Figure 3G, K and O and in Figure EV3 panels A, B, F, J and N.

2. The Authors have not addressed part of Comment 3 by Reviewer 1 regarding distinguishing between the effect of shDHRC7 on tumor engraftment versus tumor proliferation and growth. I understand that generating an inducible shRNA construct to test this could take a long time and therefore, suggest that the Authors add a sentence saying something to the effect of "not having tested to distinguish between tumor engraftment versus tumor proliferation and growth in the xenograft experiments".

Thank you for the good suggestion, we have now added this comment in lines 196-199.

3. For Comment 4 by Reviewer 2, the Authors mention "From the GSEA results, one can see that almost all E2F target genes have been downregulated (each black bar represents one gene and they are mostly on the blue (=downregulated) side of the plot)". Which GSEA results are the Authors referring to? Clarify the figure and panel, along with the text line numbers.

This refers to the Figure 4D-I.

4. For Comment 6 by Reviewer 2, the Authors mention "HMGCR expression is also elevated in RMS cells and downregulated by PROX1 silencing. We have clarified this in the revised manuscript to explain the rationale behind examining both enzymes (page 5)". Specify the line numbers in the revised manuscript which addresses this.

Lines 146-150

5. For Comment 7 by Reviewer 2, the Authors respond "This is an excellent suggestion. We are currently generating a DHCR7 overexpression construct and plan to perform these experiments. Unfortunately, this data is not available within the timeframe for the

current manuscript revision. We have discussed the role of DHCR7 in the manuscript in the context of external cholesterol supplementation, which could not rescue the phenotype. Furthermore, DHCR7 silencing induced the compensatory expression of the upstream genes/proteins in the pathway, which cannot not improve the situation either when DHCR7 levels are low. We believe that fully functional MVA pathway is thus needed for RMS cells and by adding DHCR7 back would rescue the phenotype". Are the Authors saying there is improvement with "Cannot not improve"? If the DHCR7 construct is not ready, can the Authors discuss this query raised by the Reviewer and its significance in the Discussion?

We have discussed this in lines 293-309 and modified it further in lines 306-307.

27th Oct 2025

Dear Prof. Kivelä,

We are pleased to inform you that your manuscript is accepted for publication and is now being sent to our publisher to be included in the next available issue of EMBO Molecular Medicine.

Zeljko Durdevic
Senior Editor
EMBO Molecular Medicine

Rev_Com_number: RC-2024-02836

New_manu_number: EMM-2025-21866-V3

Corr_author: Kivelä

Title: Inhibiting cholesterol synthesis halts rhabdomyosarcoma growth via ER stress and cell cycle arrest